# THE HIDDEN CONVEX OPTIMIZATION LANDSCAPE OF REGULARIZED TWO-LAYER RELU NETWORKS: AN EXACT CHARACTERIZATION OF OPTIMAL SOLUTIONS

**Yifei Wang**[*]
Department of Electrical Engineering
Stanford University
wangyf18@stanford.edu

**Jonathan Lacotte**[*]
Department of Electrical Engineering
Stanford University
lacotte@stanford.edu

**Mert Pilanci**
Department of Electrical Engineering
Stanford University
pilanci@stanford.edu

## ABSTRACT

We prove that finding all globally optimal two-layer ReLU neural networks can be performed by solving a convex optimization program with cone constraints. Our analysis is novel, characterizes all optimal solutions, and does not leverage duality-based analysis which was recently used to lift neural network training into convex spaces. Given the set of solutions of our convex optimization program, we show how to construct exactly the entire set of optimal neural networks. We provide a detailed characterization of this optimal set and its invariant transformations. As additional consequences of our convex perspective, (i) we establish that Clarke stationary points found by stochastic gradient descent correspond to the global optimum of a subsampled convex problem (ii) we provide a polynomial-time algorithm for checking if a neural network is a global minimum of the training loss (iii) we provide an explicit construction of a continuous path between any neural network and the global minimum of its sublevel set and (iv) characterize the minimal size of the hidden layer so that the neural network optimization landscape has no spurious valleys. Overall, we provide a rich framework for studying the landscape of neural network training loss through convexity.

## 1 INTRODUCTION

Let $X \in \mathbb{R}^{n \times d}$ and $y \in \mathbb{R}^n$ be the data matrix and the label vector. Given a number of neurons $m \geqslant 1$ and a regularization parameter $\beta > 0$, we consider the regularized optimization problem

$$\mathcal{P}_m^* = \min_{\theta \in \Theta_m} \left\{ \mathcal{L}_\beta(\theta) := \ell\left( \sum_{i=1}^m \sigma(Xu_i)\alpha_i \right) + \frac{\beta}{2} \sum_{i=1}^m \left( \|u_i\|_2^2 + \alpha_i^2 \right) \right\}. \tag{1}$$

where $\Theta_m = \mathbb{R}^{d \times m} \times \mathbb{R}^m$, $\theta = (U, \alpha)$, $u_i$ is the $i$-th column of $U \in \mathbb{R}^{d \times m}$ and $\alpha_i$ is the $i$-th coefficient of $\alpha \in \mathbb{R}^m$. Here we focus on the ReLU activation, i.e., $\sigma(z) = \max\{z, 0\}$ and absorb the label $y \in \mathbb{R}^n$ in the loss function $\ell : \mathbb{R}^n \to \mathbb{R}$, which is assumed to be convex (e.g., logistic, hinge, squared loss). The model $\sum_{i=1}^m \sigma(Xu_i)\alpha_i$ in (1) can be easily extended to the one with bias term by adding a column of 1's into the data $X$. We refer to an element $\theta \in \Theta_m$ as a neural network and to each pair $(u_i, \alpha_i)$ as a neuron. We denote the set of optimal neural network as

$$\Theta_m^* = \{\theta \in \Theta_m \mid \mathcal{L}_\beta(\theta) = \mathcal{P}_m^*\}. \tag{2}$$

We denote the best training loss achievable by a neural as $\mathcal{P}^* = \inf_{m \geqslant 1} \mathcal{P}_m^*$.

---

[*]Equal contributions

The ReLU activation induces a natural partition of the parameter space. We denote $D_1, \ldots, D_p$ as all possible values of $\mathrm{diag}(\mathbf{1}(Xu \geqslant 0))$. We introduce the corresponding convex cones $C_i = \{u \in \mathbb{R}^d | (2D_i - I)Xu \geqslant 0\}$ for $i \in [p]$ where we denote $[p] = \{1, \ldots, p\}$. From this partition we have the local linearization

$$\sigma(Xu) = D_i Xu, \qquad \text{for } u \in C_i. \tag{3}$$

We let $D_{i+p} = -D_i$ for $i \in [p]$ and $C_{i+p} = C_i$ for $i \in [p]$.

Such a partition of the parameter space has regained attention in the recent literature. In fact, Pilanci & Ergen (2020) recently showed that an optimal neural network $\theta^* \in \Theta_m$ for any $m \geqslant 2p$ can be constructed based on a solution of the convex optimization problem

$$\mathcal{P}_c^* := \min_{W \in \mathcal{W}} \left\{ \mathcal{L}_\beta^c(W) := \ell\Big( \sum_{i=1}^{2p} D_i Xw_i \Big) + \beta \cdot \sum_{i=1}^{2p} \|w_i\|_2 \right\}, \tag{4}$$

where we introduced the convex feasible set $\mathcal{W} := \{W = (w_1, \ldots, w_{2p}) \mid w_i \in C_i\}$. In a nutshell, this equivalence can be intuitively explained as follows. From the constraint $w_i \in C_i$, we obtain the local linearization $\sigma(Xw_i) = D_i Xw_i$ for $i \in [p]$ and $\sigma(Xw_i) = -D_i Xw_i$ for $i \in [p+1, 2p]$. By choosing neurons $(u_i, \alpha_i)$ such that $w_i = |\alpha_i| u_i$, $\alpha_i \geqslant 0$ for $i \in [p]$ and $\alpha_i \leqslant 0$ for $i \geqslant p + 1$, we further obtain by positive homogeneity of the ReLU that

$$\sum_{i=1}^{2p} D_i Xw_i = \sum_{i=1}^{2p} \sigma(Xu_i)\alpha_i. \tag{5}$$

From the fact that the cones $C_1, \ldots, C_p$ cover the entire space, Pilanci & Ergen (2020) establish the equality $\mathcal{P}_c^* = \mathcal{P}^*$, and show that an optimal neural network can be constructed from an optimal solution $w_1^*, \ldots, w_p^*$.

In this work, we explore the mapping from the optimal set of solutions $\mathcal{W}^*$ of the convex program (4) to the set of optimal neural networks $\Theta_m^*$. Our main contribution is to show how to construct the set $\Theta_m^*$ given $\mathcal{W}^*$ through simple transformations. We unveil some novel necessary conditions for a neural network to be optimal and we illustrate the relevance of these conditions by relating them to usual necessary conditions for optimality (e.g., Clarke stationarity).

## 1.1 PRIOR AND RELATED WORK

Several recent works considered over-parameterized neural networks in the infinite-width limit. In particular, it is known that in this regime, gradient descent converges to an optimal solution, see (Jacot et al., 2018; Du et al., 2018; Allen-Zhu et al., 2018; Nguyen, 2021). Further analysis in (Chizat & Bach, 2018) showed that almost no hidden neurons move from their initial values to actively learn useful features, so that this regime resembles that of kernel training and the infinite-width limit infuses convexity. Wang & Lin (2021) showed that with an explicit regularizer based on the scaled variation norm, overparametrization is generally harmless to two-layer ReLU networks. However, experiments in (Arora et al., 2016) suggest that this kernel approximation is unable to fully explain the success of non-convex neural network models.

Convexity arguments in neural networks were proposed in the recent literature (Bengio et al., 2006; Bach, 2017). However, existing works, except (Pilanci & Ergen, 2020), are restricted to infinitely wide networks. In turn, Bengio et al. (2006) and Bach (2017) consider greedy neuron-wise optimization strategies for the infinite-dimensional optimization problem, which requires solving non-convex problems at every step to train a shallow neural network. In contrast, in our work, we reveal the hidden convex optimization landscape for any finite number of hidden neurons.

Besides the convexity properties of infinitely wide networks, many works derived lower bounds on the hidden layer size to guarantee the absence of spurious minima. Venturi et al. (2019) showed that the *un-regularized* (i.e. $\beta = 0$) objective $\mathcal{L}_\beta$ has no spurious local minima provided that the number of neurons satisfies $m \geqslant n$, and a similar result was shown in (Livni et al., 2014). Similar results were derived for deep networks. For instance, Soudry & Carmon (2016) showed that under a dropout-like noise assumption, there exist no differentiable spurious minima if the product of the dimensions of the layer weights exceeds $n$ and this result matches the classical lower bound (Baum, 1988) on the minimal width of a neural network to implement any dichotomy for inputs in general position. In a

similar vein, Nguyen & Hein (2017) showed that no spurious minima occur provided that one of the layer's inner width exceeds $n$ and under additional non-degeneracy conditions. For activations other than the ReLU (e.g., linear, quadratic, polynomial), similar lower bounds were derived in (Venturi et al., 2019; Du & Lee, 2018; Soltanolkotabi et al., 2018). These analyses are typically based on the idea that when $m \gtrsim n$ then it is very likely that the features $\sigma(Xu_1), \ldots, \sigma(Xu_m)$ form a basis of $\mathbb{R}^n$ so that the training problem reduces to finding a linear model with weights $\alpha_1, \ldots, \alpha_m$ which perfectly fits the labels. For the hinge loss and linear separable data, (Wang et al., 2019) show that the modified stochastic gradient descent method can achieve global optimality despite the presence of spurious local minima and saddle points.

The training landscape of neural networks is of great interest for theoretical analysis in the optimization of neural networks. An important perspective is to analyze the landscape via paths through the parameter space, see (Vidal et al., 2017). Indeed, in (Haeffele & Vidal, 2015; 2017; Sharifnassab et al., 2019), it is shown that there exists a non-increasing path in objective value from every point to the global minimum with mild assumption on the layer width. The existence of such paths also indicates that the level sets of the training loss are connected (Freeman & Bruna, 2016; Venturi et al., 2019; Nguyen, 2019; Nguyen et al., 2021) and there is no bad local valley (Nguyen & Hein, 2017).

However, with regularization, the training problem is more challenging. Intuitively, it reduces the set of optimal solutions to those with small norms. Without regularization (i.e., $\beta = 0$), it should be noted that the set of optimal solutions always contains infinitely many points. For instance, with ReLU activations, it holds that if $\theta^* = \{(u_i^*, \alpha_i^*)\}_{i=1}^m$ is an optimal neural network, then any re-scaling of $\theta^*$ in the form $\{(\frac{u_i^*}{\gamma_i}, \gamma_i \alpha_i^*)\}_{i=1}^m$ (with $\gamma_1, \ldots, \gamma_m > 0$) has the same objective value and is thus optimal. With regularization, this manifold is reduced to a single point. It is then natural to expect that this minimal size of the hidden layer must increase. Further, the aforementioned analyses do not extend since regularization also penalizes the norms of the $u_i$'s, and one cannot simply generate such a basis of $\mathbb{R}^n$ based on the features $\sigma(Xu_1), \ldots, \sigma(Xu_m)$ by random sampling and then overfitting the labels.

Recently, Pilanci & Ergen (2020); Ergen & Pilanci (2020) show that two-layer ReLU neural networks can be optimized exactly via finite-dimensional convex programs with complexity polynomial in the number of samples and hidden neurons. As indicated in Pilanci & Ergen (2020), the worst-case complexity is exponential in the dimension of the training samples unless $P = NP$.

## 1.2 SUMMARY OF OUR CONTRIBUTIONS

In Section 2, we introduce the notions of minimal neural networks and nearly minimal neural networks. These two notions are closely related to the plateau and the edge of the plateau of the loss landscape.

In Section 3, we show that any minimal neural network $\theta$ can be represented, via an explicit map, in the convex feasible space $\mathcal{W}$ as a point $W(\theta)$ such that $\mathcal{L}_\beta(\theta) \geqslant \mathcal{L}_\beta^c(W(\theta))$, and vice-versa. This structural result provides a mathematically rich perspective to characterize optimal neural networks through the lens of convexity. We then provide an exact characterization of the set of all global optima of the nonconvex problem, which include all nearly minimal neural networks generated via the optimal solutions of the convex program.

In Section 4, we show that any Clarke stationary point $\theta$ with respect to $\mathcal{L}_\beta$ is a nearly minimal neural network. This provides a preliminary structure on the solutions found by stochastic gradient descent (SGD), as it has been recently shown (see, for instance, Corollary 5.11 in Davis et al. (2020)) that the limit points of SGD applied to neural network optimization are Clarke stationary. More importantly, we show that Clarke stationary point $\theta$ with respect to $\mathcal{L}_\beta$ also corresponds to a global minimum of a subsampled convex problem. We also provide a polynomial-time algorithm (in the sample size $n$ and the hidden-layer size $m$) in order to test whether a neural network is globally optimal.

In Section 5, we show that any neural network is path-connected to a succinct representation (with at most $n + 1$ non-zero neurons) and this path is with constant objective value. Then, from the convex perspective of two-layer ReLU neural networks, we provide an explicit path of non-increasing loss between $\theta$ and $\theta'$, where $\theta'$ is the global optimum of the non-convex training problem. This establishes that the training loss $\mathcal{L}_\beta$ has no spurious local minima, provided that the number of neurons is sufficiently large.

## 1.3 NOTATIONS

We first present an alternative interpretation of the cones $C_i$ and the diagonal matrices $D_i$ for $i \in [p]$. The ReLU activation function partitions the space of neurons $u \in \mathbb{R}^d$ into linearly separated regions, that is, given a binary vector $s \in \{0,1\}^n$, the set of neurons $u \in \mathbb{R}^d$ such that $\mathbf{1}(Xu \geqslant 0) = s$ is a convex cone in $\mathbb{R}^d$, if not empty. We enumerate the closures of all these cones as $C_1, \ldots, C_p$ and we set $C_{i+p} = C_i$ for $i \in [p]$. For $i \in [p]$, we introduce the corresponding diagonal matrices $D_i = \mathrm{diag}(\mathbf{1}(Xu \geqslant 0))$ for an arbitrary $u \in C_i$, and $D_{i+p} = -D_i$. Here the number $p$ is the number of dichotomies that the data matrix $X$ can realize. It is upper bounded by $p \leqslant 2r \left( \frac{e(n-1)}{r} \right)^r$ where $r = \mathrm{rank}(X)$, see (Cover, 1965).

Beyond the dichotomies of the space of neurons $u \in \mathbb{R}^d$, we further introduce the partitions (trichotomies) $\{I_+, I_0, I_-\}$ of $[n]$ such that there exists a solution vector $u \in \mathbb{R}^d$ verifying $(Xu)_k > 0$ if $k \in I_+$, $(Xu)_k = 0$ if $k \in I_0$ and $(Xu)_k < 0$ if $k \in I_-$. Clearly, there exists a finite number $q$ of such trichotomies and $q$ is trivially upper bounded by $3^n$. For the $j$-th trichotomy $\{I_+, I_0, I_-\}$, we define the $n \times n$ diagonal matrix $T_j$ with $k$-th diagonal element $(T_j)_{kk} = 1$ if $k \in I_+$, $(T_j)_{kk} = 0$ if $k \in I_0$ and $(T_j)_{kk} = 0$ if $k \in I_-$. Such trichotomies are also discussed in Phuong & Lampert (2020).

For each $j = 1, \ldots, q$, we define $Q_j$ as the *closed convex cone* of solution vectors for the $j$-th trichotomy $\{I_+, I_0, I_-\}$. We consider a partition $\{B_1, \ldots, B_{2q}\}$ of the neurons' parameter space where $B_i := Q_i \times \mathbb{R}_{>0}$ for $j = 1, \ldots, q$ and $B_j := Q_{j-q} \times \mathbb{R}_{<0}$ for $j = q+1, \ldots, 2q$. We augment the set of diagonal matrices $\{T_j\}_{j=1}^q$ by setting $T_j = -T_{j-q}$ for $j = q+1, \ldots, 2q$.

For a neuron pair $(u, \alpha) \in \mathbb{R}^d \times \mathbb{R}$, we denote $B(u, \alpha)$ as the unique $B_i$ such that $(u, \alpha) \in B_i$.

The notion of path-connected sublevel set is introduced as follows.

**Definition 1.** *We write $\theta \blacktriangleright \theta'$ if the neural network $\theta' \in \Theta_m$ belongs to the path-connected sublevel set (or valley) of $\theta \in \Theta_m$. Namely, there exists a continuous path $\gamma : [0,1] \to \Theta_m$ such that $\gamma(0) = \theta$, $\gamma(1) = \theta'$ and $t \mapsto \mathcal{L}_\beta(\gamma(t))$ is non-increasing. We denote the valley of $\theta$ as $\Omega(\theta) := \{\theta' \in \Theta_m \mid \theta \blacktriangleright \theta'\}$. We say that $\theta \in \Theta_m$ is non-spurious if $\theta \blacktriangleright \theta^*$ for some $\theta^* \in \mathrm{argmin}_{\theta' \in \Theta_m} \mathcal{L}_\beta(\theta')$. Otherwise, we say that $\theta$ and its valley $\Omega(\theta)$ are spurious.*

## 2 MINIMAL NEURAL NETWORKS AND NEARLY MINIMAL NEURAL NETWORKS

We start with the notion of minimal neural networks and nearly minimal neural networks. Minimal neural networks enjoy a well-structured representation which is useful to understand the optimality properties of two-layer neural networks.

**Definition 2** (Minimal neural networks). *We say that a neural network $\theta$ is minimal if (i) it is scaled, i.e., $\|u_i\|_2 = |\alpha_i|$ for $i \in [m]$ and (ii) the cones $B(u, \alpha)$ of each of its non-zero neurons $(u, \alpha)$ are pairwise distinct. That is, a minimal neural network has at most a single non-zero neuron per cone $B_i$. We denote by $\Theta_m^{min}$ the set of minimal neural networks with $m$ neurons.*

Note that any minimal neural network has at most $2q$ non-zero neurons since there are $2q$ cones $B_i$ and at most one neuron per cone. Next, we introduce a slightly less structured class of neural networks that one can interpret as 'split' versions of minimal neural networks, and can have an arbitrary number of non-zero neurons.

**Definition 3** (Nearly minimal neural networks). *We say that a neural network $\theta$ is nearly minimal if (i) it is scaled and (ii) for any two non-zero neurons $(u, \alpha)$, $(v, \beta)$ of $\theta$, if $B(u, \alpha) = B(v, \beta)$ then $u$ and $v$ are positively colinear, i.e., there exists $\lambda \geqslant 0$ such that $u = \lambda v$. We denote by $\widetilde{\Theta}_m^{min}$ the set of nearly minimal neural networks with $m$ neurons. It trivially holds that $\Theta_m^{min} \subset \widetilde{\Theta}_m^{min}$.*

For a nearly minimal neural network, by merging the neurons corresponding to the same trichotomies, we can reformulate it into a minimal neural network without changing the objective value.

### 2.1 FROM NEARLY MINIMAL TO MINIMAL NEURAL NETWORKS

Nearly minimal neural networks have the property that any two neurons which share at least one active cone must be positively colinear. As we establish next, these colinear neurons can be continuously merged together along a path of constant objective value, resulting in a minimal neural network.

Formally, we let $\theta \in \widetilde{\Theta}_m^{\min}$ be a nearly minimal neural network with $m$ neurons and we fix $(w, \gamma) \in \theta$ a non-zero neuron. Let $(w_2, \gamma_2), \ldots, (w_k, \gamma_k) \in \theta$ be the other non-zero neurons such that for each $j = 2, \ldots, k$, we have $\mathrm{sign}(\gamma) = \mathrm{sign}(\gamma_j)$, and, $w$ and $w_j$ are positively colinear. Write $(w_1, \gamma_1) := (w, \gamma)$, and define the merged neuron $(w^m, \gamma^m)$ as $w^m := \frac{\sum_{j=1}^k |\gamma_j| w_j}{\sqrt{\| \sum_{j=1}^k |\gamma_j| w_j \|_2}}$ and $\gamma^m := \mathrm{sign}(\gamma) \| w^m \|_2$. Let $\mathcal{M}(\theta)$ be a copy of $\theta$ where each such set of $k$ positively colinear neurons $\{(w_1, \gamma_1), \ldots, (w_k, \gamma_k)\}$ is replaced by the $k$ neurons $\{(w^m, \gamma^m), (0, 0), \ldots, (0, 0)\}$. We refer to $\mathcal{M}(\theta)$ as the *merged* version of $\theta$. The next result states relevant properties of $\mathcal{M}(\theta)$.

**Proposition 1.** *Let $\theta \in \widetilde{\Theta}_m^{\min}$. Then, the following results hold.*

1. *The merged neural network $\mathcal{M}(\theta)$ is a minimal neural network.*
2. *We have $\theta \blacktriangleright \mathcal{M}(\theta)$, and the continuous path from $\theta$ to $\mathcal{M}(\theta)$ has constant objective value.*
3. *If $\mathcal{M}(\theta)$ is a local minimum of $\mathcal{L}_\beta$, then $\theta$ is also a local minimum of $\mathcal{L}_\beta$.*

Intuitively, merging the colinear neurons preserves the active cones and leaves a single neuron per cone, so that $\mathcal{M}(\theta)$ is indeed minimal. The third property essentially follows from the fact that $\mathcal{M}(\theta)$ has more degrees of freedom than $\theta$ since it has more neurons equal to $0$. In addition, the continuous path of constant objective value from $\theta$ to $\mathcal{M}(\theta)$ can be explicitly constructed (see the proof in Appendix B.1).

## 3 Mapping neural networks to a convex optimization landscape

We provide here an explicit map from the set of minimal neural networks to the feasible set $\mathcal{W}$ of the convex program (4), and vice-versa. For $W = (w_1, \ldots, w_{2p}) \in \mathcal{W}$, we let $\| W \|_0$ be number of the non-zero vectors in $w_1, \ldots, w_{2p}$. Define

$$\mathcal{W}_m = \{ W \in \mathcal{W} \mid \| W \|_0 \leqslant m \}, \qquad \mathcal{W}_m^* = \mathcal{W}_m \cap \mathcal{W}^*. \tag{6}$$

First, we introduce the map $\theta \mapsto W(\theta)$ from $\Theta_m^{\min}$ to $\mathcal{W}_m$ where for each $i = 1, \ldots, 2p$, we set

$$w_i(\theta) := \sum_{\substack{j=1,\ldots,m \\ B(u_j, \alpha_j) \subseteq B_i}} |\alpha_j| u_j, \tag{7}$$

and such that each non-zero neuron $(u_j, \alpha_j)$ contributes only to a single $w_i$. To understand the latter, note that each cone $B(u_j, \alpha_j)$ might be a subset of several (adjacent) cones $B_i$ and hence, one might need to choose which $w_i$ a neuron $(u_j, \alpha_j)$ contributes to. These ties can be resolved arbitrarily without affecting any of our results.

Conversely, we construct a map $W \mapsto \theta(W)$ from $\mathcal{W}_m$ to $\Theta_m^{\min}$ by setting $\theta(W) = \{(u_i, \alpha_i)\}_{i=1}^m$ where the $(u_i, \alpha_i)$ are defined as follows. Denote $i_1 < \cdots < i_m$ the indices such that if $i \notin \{i_1, \ldots, i_m\}$ then $w_i = 0$. Take the index $J$ (if any) such that $i_J \leqslant p$ and $i_{J+1} \geqslant p + 1$. Let $\{K_1, \ldots, K_\ell\}$ be a partition of $\{i_1, \ldots, i_J\}$ in terms of the repartition of $w_{i_1}, \ldots, w_{i_J}$ into the cones $\{Q_1, \ldots Q_q\}$. Similarly, let $\{K_{\ell+1}, \ldots, K_{\ell+\ell'}\}$ be a partition of $\{i_{J+1}, \ldots, i_m\}$ in terms of the repartition of $w_{i_{J+1}}, \ldots, w_{i_m}$ into the cones $\{Q_1, \ldots Q_q\}$. Then, for $1 \leqslant j \leqslant \ell + \ell'$, we set

$$(u_j, \alpha_j) := \left( \frac{\sum_{i \in K_j} w_i}{\sqrt{\| \sum_{i \in K_j} w_i \|_2}}, \ \gamma_j \cdot \sqrt{\| \sum_{i \in K_j} w_i \|_2} \right), \tag{8}$$

where $\gamma_j = 1$ if $j \leqslant \ell$ and $\gamma_j = -1$ if $j > \ell$. Finally, for $\ell + \ell' + 1 \leqslant j \leqslant m$, we set $(u_j, \alpha_j) = (0, 0)$. As stated in the next result, these mappings can only improve the training loss.

**Proposition 2.** *It holds that for any $W \in \mathcal{W}_m$ we have $\mathcal{L}_\beta(\theta(W)) \leqslant \mathcal{L}_\beta^c(W)$, and, for any $\widetilde{\theta} \in \Theta_m^{\min}$, we have $\mathcal{L}_\beta^c(W(\widetilde{\theta})) \leqslant \mathcal{L}_\beta(\widetilde{\theta})$. Furthermore, it holds that $\theta(W(\widetilde{\theta})) \in \Omega(\widetilde{\theta})$.*

These mappings between minimal neural networks and the convex feasible set provide a rich structure to address the optimality properties of neural networks. In Figure 1, we provide an illustration of the non-convex and convex landscapes on a toy neural network training model.

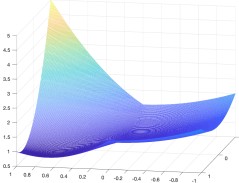 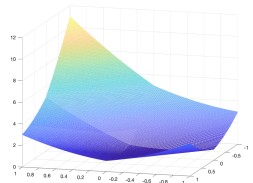

Figure 1: Comparison of the non-convex landscape (left) and the convex landscape (right) of program (4). Here, we consider the toy example with date $X = 1$, label $y = 1$ and the $\ell_2$ loss. Then, we have $\mathcal{L}_\beta(u, \alpha) = (1 - \max\{u, 0\}\,\alpha)^2 + \frac{1}{2}(|u|^2 + |\alpha|^2)$. The convex objective is then $\mathcal{L}_\beta^c(v, w) = (1 - v + w)^2 + (|v| + |w|)$ subject to $v, w \geqslant 0$. The set of minimal neural networks corresponds to $|u| = |\alpha|$, which includes the optima. Further, the optimal values of the two functions match and are equal to 0.75, and attained at $(u, \alpha) = (1/\sqrt{2}, 1/\sqrt{2})$ and $(v, w) = (1/2, 0)$. Note that $u|\alpha| = v$, and this indeed corresponds to our mapping (7).

### 3.1 THE GLOBAL OPTIMAL SET OF NEURAL NETWORKS

Let $m^* = \min_{W \in \mathcal{W}^*} \|W\|_0$. As a consequence of Caratheodory's theorem, we have the following upper bound on the minimal cardinality $m^*$ of an optimal solution.

**Lemma 1.** *It holds that $m^* \leqslant n + 1$. Further, for any $m \geqslant m^*$, we have that $\mathcal{P}_m^* = \inf_{k \geqslant 1} \mathcal{P}_k^*$.*

From the definition of $m^*$, it clearly holds that $\mathcal{W}_m^* \neq \varnothing$ for $m \geqslant m^*$. Then, we present the mapping from the optimal solution to the convex problem (4) to a globally optimal neural network for the non-convex problem (1).

**Lemma 2.** *Let $W = (w_1, \ldots, w_{2p}) \in \mathcal{W}^*$, and denote by $\mathcal{I} = \{i_1, \ldots, i_{\|W\|_0}\} \subset [2p]$ the set of indices such that $w_i^* \neq 0$ for $i \in \mathcal{I}$. We set*

$$(u_j, \alpha_j) = \left( \frac{w_{i_j}}{\sqrt{\|w_{i_j}\|_2}}, \gamma_{i_j} \sqrt{\|w_{i_j}\|_2} \right), \tag{9}$$

*for $i_j \in \mathcal{I}$. Here $\gamma_i = 1$ if $i \leqslant p$ and $\gamma_i = -1$ if $i > p$. Then, it holds that $\theta = \{(u_i, \alpha_i)\}_{i=1}^{\|W\|_0}$ is an optimal neural network, i.e., $\mathcal{L}_\beta(\theta) = \mathcal{P}^*$.*

We denote the above mapping (9) by $\psi$, and we set $\Theta_m^{\mathrm{cvx}} = \psi(\mathcal{W}_m^*)$. According to Lemma 2, it holds that $\Theta_m^{\mathrm{cvx}} \subseteq \Theta_m^*$. Given a neuron $(u, \alpha)$, we say that a collection of neurons $\{(u_j, \alpha_j)\}_{j=1}^k$ is a splitting of $(u, \alpha)$ if $(u_j, \alpha_j) = (\sqrt{\gamma_j} u, \sqrt{\gamma_j} \alpha)$ for some $\gamma_j \geqslant 0$ and $\sum_{j=1}^k \gamma_j = 1$. Given a neural network $\theta = \{(u_i, \alpha_i)\}_{i=1}^m$, a *splitting* of $\theta$ is any neural network $\theta' \in \Theta_m$ such that the non-zero neurons of $\theta'$ can be partitioned into splittings of the neurons of $\theta$. Similarly, split neurons can be *merged* back to their original form. We denote by $\tilde{\Theta}_m^{\mathrm{cvx}}$ the set of splittings generated from $\Theta_m^{\mathrm{cvx}}$. We provide an exact characterization of the optimal set in the following theorem.

**Theorem 1.** *Suppose that $m \geqslant m^*$. It holds that $\Theta_m^* = \tilde{\Theta}_m^{\mathrm{cvx}}$. Namely, all optimal solutions of the nonconvex loss can be found via the optimal solutions of the convex program (4) up to permutation and splitting/merging of the neurons as defined above.*

We compare our result with the result in (Pilanci & Ergen, 2020) as follows. Essentially, Pilanci & Ergen (2020) show how to construct *one* globally optimal solution of the nonconvex loss by solving the convex program, while Theorem 1 shows how to construct the *entire* set of global optimum of the nonconvex loss. The relations among $\mathcal{W}_m^*, \Theta_m^{\mathrm{cvx}}, \tilde{\Theta}_m^{\mathrm{cvx}}$ and $\Theta_m^*$ is illustrated in Figure 2.

**Example 1.** *We consider a toy example, where $X = \begin{bmatrix} 1 & 0 \\ 0 & 1 \\ 1 & 1 \end{bmatrix}$, $Y = \begin{bmatrix} 1 \\ 0 \\ 0 \end{bmatrix}$ and $\beta = 0.1$. In this case, $p = 6$ and we can enumerate the diagonal matrices $D_i$ as*

$$\begin{aligned} &D_1 = \mathrm{diag}([0, 0, 0]), D_2 = \mathrm{diag}([0, 1, 0]), D_3 = \mathrm{diag}([0, 1, 1]), \\ &D_4 = \mathrm{diag}([1, 0, 0]), D_5 = \mathrm{diag}([1, 0, 1]), D_6 = \mathrm{diag}([1, 1, 1]). \end{aligned} \tag{10}$$

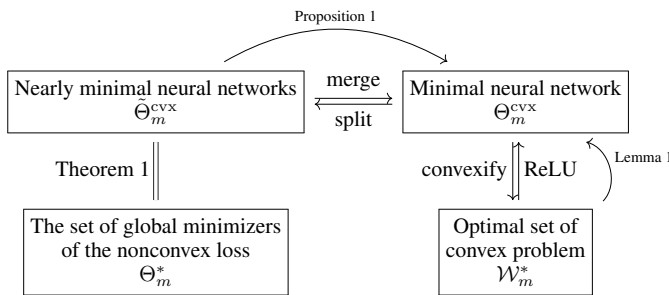

Figure 2: Illustration of relations between $\mathcal{W}_m^*, \Theta_m^{\text{cvx}}, \tilde{\Theta}_m^{\text{cvx}}$ and $\Theta_m^*$.

*The optimal solution to the convex problem* (4) *is given by* $W^* = (w_1^*, \ldots, w_{2p}^*)$, *where* $W^*$ *only consists of one non-zero block* $w_5^* = [0.86, -0.79]^T$. *Therefore, the set of the global minimizers of the nonconvex loss* $\mathcal{L}_\beta$ *consists of all nearly minimal neural network* $\theta = \{(u_i, \alpha_i)\}_{i=1}^m$ *satisfying*

$$u_i = \sqrt{\gamma_i} \frac{w_5^*}{\sqrt{\|w_5^*\|_2}}, \quad \alpha_i = \sqrt{\gamma_i} \sqrt{\|w_5^*\|_2}, \qquad i \in [m], \tag{11}$$

*where* $\sum_{i=1}^m \gamma_i = 1$ *and* $\gamma_i \geqslant 0$ *are arbitrary. These correspond to the split versions of the single neuron* $w_5^*$. *We investigate numerically our result: for* $m = 5$, *we run gradient descent (GD) on the nonconvex loss* $\mathcal{L}_\beta$ *until we find a nearly stationary neural network* $\{(u_i, \alpha_i)\}_{i=1}^5$. *We plot the points* $\alpha_i u_i$ *as well as* $w_5^*$ *in Figure 3 in the Appendix.*

## 4 CHARACTERIZATION OF ALL LOCAL MINIMA

Minimal neural networks form a subset considerably smaller than the entire space of neural networks, and they do contain all the global optima. In this section, we show that first-order methods can find networks that can be merged to a minimal representation. Moreover, we exhibit the existence of a path of strictly decreasing objective value from any neural network to a minimal representation. This may suggest that minimal neural networks are the right notion to study the complexity of the loss landscape.

### 4.1 SGD FINDS A NEARLY MINIMAL NEURAL NETWORK

The limit points of SGD are almost surely Clarke stationary with respect to $\mathcal{L}_\beta$ (see, e.g., (Davis et al., 2020; Bolte & Pauwels, 2019)). We show next that any Clarke stationary point w.r.t. the loss $\mathcal{L}(\theta)$ is in fact a nearly minimal neural network. This shows that SGD finds a neural network which can be merged to a minimal representation.

**Theorem 2.** *Fix* $m \geqslant 1$. *Any Clarke stationary point* $\theta$ *of the non-convex loss function* $\mathcal{L}_\beta$ *over* $\Theta_m$ *is a nearly minimal neural network. Consequently, any local minimum of* $\mathcal{L}_\beta$ *is nearly minimal.*

As an additional motivation for studying nearly minimal neural networks, we establish the following.

**Proposition 3.** *Let* $\theta \in \Theta_m$ *be any neural network. There exists a continuous path in* $\Theta_m$ *from* $\theta$ *to a nearly minimal neural network along which the loss function is (strictly) decreasing.*

The proof of Theorem 2 is deferred to Appendix B.4 and that of Proposition 3 to Appendix B.5. Both proofs are based on the same transformations of a neural network $\theta$ which decreases the training loss: scaling the neural network and then aligning the non-zero neurons which belong to the same cones $B_i$ so that they become positively colinear. These transformations leave the predictions unchanged due to the piecewise linear structure of the activation function but decrease the value of the regularization term. Thus, our notions of minimal representations are intimately related to (i) the piecewise linear structure of the activation function and (ii) the regularization effect. We emphasize again that these two features of neural network training are commonly used in practice (e.g., ReLU and weight decay).

Combining Proposition 3 and Proposition 1, we immediately obtain the following result.

**Corollary 1.** *The valley $\Omega(\theta)$ of any neural network $\theta$ contains a minimal one. Further, if the valley $\Omega(\theta)$ is non-spurious, then it contains an optimal neural network which is minimal.*

Interestingly, we are able to provide an explicit construction of the map from a neural network $\theta$ to a nearly minimal representation, and this map is based on the aforementioned transformations (scaling and aligning; see the proof of Proposition 3 for details).

Hence, the study of the optimality properties of a neural network can be narrowed down to the structured class $\widetilde{\Theta}_m^{\min}$ which contains the limit points of SGD. Next, we establish that we can go further by considering the class of minimal neural networks $\Theta_m^{\min}$.

### 4.2 CLARKE'S STATIONARY POINT AND SUBSAMPLED CONVEX PROGRAM

Consider the convex program with trichotomies:

$$\min \ \ell\Big( \sum_{j=1}^{q} T_j X(w_j - w_{j+q}) \Big) + \beta \sum_{j=1}^{2q} \|w_j\|_2, \quad \text{s.t. } w_j, w_{j+q} \in Q_j, j \in [q]. \tag{12}$$

The convex program with trichotomies also provides a convex optimization formulation of the regularized neural network training problem (1).

**Proposition 4.** *The convex program* (12) *with trichotomies has the optimal value* $\mathcal{P}^*$.

Given a subset $\mathcal{I} \subseteq [q]$, we can also consider a subsampled convex program with trichotomies:

$$\min \ \ell\Big( \sum_{j \in \mathcal{I}} T_j X(w_j - w_{j+q}) \Big) + \beta \sum_{j \in \mathcal{I}} (\|w_j\|_2 + \|w_{j+q}\|_2), \quad \text{s.t. } w_j, w_{j+q} \in Q_j, j \in \mathcal{I}. \tag{13}$$

We show the connection of the Clarke's stationary point of the nonconvex loss function $\mathcal{L}_\beta$ and the optimal solution of the subsampled convex program (13) as follows.

**Theorem 3.** *Suppose that $\theta$ is a Clarke's stationary point of the nonconvex loss function $\mathcal{L}_\beta$. Let $\mathcal{I} = \{j \in [q] \mid \text{there exists } k \in [m] \text{ such that } T_j = \text{diag}(\text{sign}(Xu_k))\}$. Then, $\theta$ corresponds to a global optimum of the subsampled convex program* (13).

In other words, any local minimum of the nonconvex loss (1) can be characterized as a global minimum of a subsampled convex program (13); further, the optimality gap is equal to the gap between the subsampled problem (13) and the full convex program (12).

### 4.3 SUBSAMPLED CONVEX PROGRAM AND VERIFYING GLOBAL OPTIMALITY

We established that a stationary point of the non-convex training loss is a global optimum of a subsampled convex program. Here, we build on this observation to design a procedure to check whether a neural network is in fact a global minimizer. Our key theoretical contribution is to provide such an algorithm that runs in polynomial time of sample size $n$.

We first note that the set $\{D_i\}_{i=1}^p$ can be constructed in polynomial time of $n$ via standard results from geometry and hyperplane arrangements in (Cover, 1965; Winder, 1966; Ojha, 2000). Consider a feasible point $W = (w_1, \ldots, w_{2p}) \in \mathcal{W}$ of the convex program (4). Note that each constraint $w_i \in C_i$ is a linear inequality constraint. Indeed, as described in Section 2, each $C_i$ is the convex cone of solution vectors for a dichotomy $\{I_+, I_-\}$ of $\{1, \ldots, n\}$. Writing $X_+^{(i)}$ (resp. $X_-^{(i)}$) the subset of rows of $X$ indexed by $I_+$ (resp. $I_-$), we have $u \in C_i$ if and only $X_+^{(i)} u \succeq 0$ and $X_-^{(i)} u \preceq 0$. Using these notations, the convex program (4) can be reformulated as

$$\min_{w_1, \ldots, w_{2p}} \ \ell\big(\widehat{y}_c(W)\big) + \beta \sum_{i=1}^{2p} \|w_i\|_2 \quad \text{s.t.} \quad X_+^{(i)} w_i \succeq 0, \ X_-^{(i)} w_i \preceq 0 \ \forall i = 1, \ldots, 2p,$$

where $\widehat{y}_c(W) = \sum_{i=1}^{2p} D_i X w_i$. Hence, given a feasible point $W^* = (w_1, \ldots, w_{2p}) \in \mathcal{W}$ to the convex program (4), it holds that $W^*$ is a global minimizer if and only if $W^*$ satisfies the Karush-Kuhn-Tucker (KKT) conditions (see (Boyd et al., 2004)) of (4). Here, $W^* \in \mathcal{W}$ satisfies the

KKT conditions if, for each $i = 1, \ldots, 2p$, there exist $\zeta_+^{(i)}, \zeta_-^{(i)} \succeq 0$ such that $\langle \zeta_+^{(i)}, X_+^{(i)} w_i^* \rangle = \langle \zeta_-^{(i)}, X_-^{(i)} w_i^* \rangle = 0$ and

$$X^\top D_i \nabla \ell(\widehat{y}_c(W^*)) + \beta \frac{w_i^*}{\|w_i^*\|_2} + X_-^{(i)\top} \zeta_-^{(i)} - X_+^{(i)\top} \zeta_+^{(i)} = 0, \qquad \text{if } w_i^* \neq 0 \qquad (14)$$

$$\left\| X^\top D_i \nabla \ell(\widehat{y}_c(W^*)) + X_-^{(i)\top} \zeta_-^{(i)} - X_+^{(i)\top} \zeta_+^{(i)} \right\|_2 \leqslant \beta, \qquad \text{otherwise}. \qquad (15)$$

This amounts to solving a system with $2np$ variables of $2np$ linear inequalities, $n_0$ convex quadratic inequalities and $(2p - n_0)(d + 2)$ linear equalities, where $n_0$ is the number of variables $w_i^*$ equal to 0, and this can be done efficiently using standard convex solvers, in time polynomial in the sample size $n$. The next result establishes the link between checking the KKT conditions of the above program and checking whether a neural network is a global optimum. Its proof is deferred to Appendix B.8.

**Proposition 5.** *Let $\widetilde{\theta} \in \Theta_m^{\min}$ be a minimal neural network. Suppose that $W(\widetilde{\theta})$ satisfies the KKT conditions as described above. Then, $\theta(W(\widetilde{\theta}))$ is a global optimum of the loss $\mathcal{L}_\beta$.*

In the above result, the minimal neural network assumption is not restrictive, since any local minima of the loss must be a nearly minimal neural network (Theorem 2), and then, any nearly minimal neural network can be reduced to a minimal one along a continuous path of constant value (Theorem 1).

## 5   NON-SPURIOUS VALLEYS AND CONVEX LANDSCAPE

The subsampled convex program relates to the optima of SGD. It is then of interest to understand the landscape when $m$ is much smaller than the number of cones. Here we show that a critical threshold is $n + 1 + m^*$ for having a path of non-increasing value. While this may be an open problem, it is reasonable to expect SGD to behave better in that case, and thus to find a global minimum. For an arbitrary neural network $\theta$, we can find a point $\theta'$ with at most $n + 1$ non-zero neurons such that they are connected with a path with constant objective values.

**Proposition 6.** *Given a scaled neural network $\theta \in \Theta_m$ with $m \geqslant n + 1$, there exists a neural network $\theta'$ with at most $n + 1$ non-zero neurons and there exists a path with constant objective value between $\theta$ and $\theta'$. Namely, $\theta \blacktriangleright \theta'$ and $\theta' \blacktriangleright \theta$.*

A direct corollary of Proposition 6 is that for any global optimum $\theta^*$ of $\mathcal{L}_\beta$, we can find a succinct representation $\widetilde{\theta}^*$ with at most $n + 1$ non-zero neurons and there exists a path between $\widetilde{\theta}^*$ and $\theta^*$ such that the objective value is constant. Based on Proposition 6, we can also show that there is no spurious valley. The following result states the absence of spurious valleys for the training loss as soon as $m \gtrsim n$.

**Proposition 7.** *Let $m \geqslant n + 1 + m^*$. Then, it holds that for any neural network $\theta \in \Theta_m$, we have $\theta \blacktriangleright \theta^*$ for some $\theta^* \in \Theta_m^*$.*

In other words, provided that $m \geqslant n + 1 + m^*$, *all strict local minima are global*. Compared to the standard lower bound $m \geqslant n$ for the unregularized case in (Venturi et al., 2019; Livni et al., 2014), we have an additional term $m^* \leqslant n + 1$ induced by weight decay.

As known in the literature (Freeman & Bruna, 2016; Venturi et al., 2019; Vidal et al., 2017), the loss landscape of an over-parameterized shallow neural network is almost convex. Essentially, for a sufficiently wide neural network, for any $\theta_0, \theta_1$ and $\lambda \in [0, 1]$, we can find $\theta_\lambda$ such that $f(x; \theta_\lambda) = \lambda f(x; \theta_0) + (1 - \lambda) f(x; \theta_1)$. From a perspective of convex formulation of two layer neural network, we give a sufficient upper bound on the width of neural network to ensure the convex landscape in terms of realizations. Essentially, as long as $m \geqslant 2(n + 1)$, for any two neural network realizations $f(x; \theta_0)$ and $f(x; \theta_0)$, we can find succinct representations $\widetilde{\theta}_1, \widetilde{\theta}_2$ such that $\theta_i \blacktriangleright \widetilde{\theta}_i$ and $\widetilde{\theta}_i \blacktriangleright \theta_i$ for $i = 1, 2$. Then, for any $\lambda \in [0, 1]$, we can construct $\theta_\lambda$ such that

$$f(x; \theta_\lambda) = \lambda f(x; \widetilde{\theta}_0) + (1 - \lambda) f(x; \widetilde{\theta}_1) \qquad (16)$$

The construction of $\theta_\lambda$ is straightforward. From Proposition 6, we can take $\widetilde{\theta}_i$ as a neural network with at most $n + 1$ non-zero neurons for $i = 1, 2$. Given $m \geqslant 2(n + 1)$, following the proof of Proposition 7, we can construct $\theta_\lambda$ satisfying (16).

## ACKNOWLEDGEMENTS

This work was partially supported by the National Science Foundation under grants ECCS-2037304, DMS-2134248, and the Army Research Office.

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

## A   FIGURE IN EXAMPLE 1

We plot the points $\alpha_i u_i$ as well as $w_5^*$ in Figure 3.

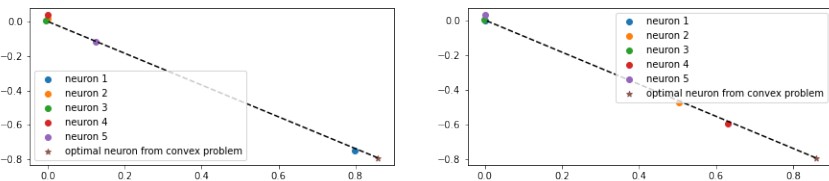

Figure 3: Plots in the neuron space of nerual networks trained by GD over two trials. Points $\alpha_i u_i$ of the neural network $\{(u_i, \alpha_i)\}_{i=1}^m$ trained by GD, and non-zero block $w_5^*$ of the global solution of the convex problem. The points $\alpha_i u_i$ lie in the convex hull of $\{0, w_5^*\}$, and they satisfy equation (11) up to numerical tolerance. This implies in particular that the neural network found by GD is optimal.

## B   PROOFS OF MAIN RESULTS

Thoughout the appendix, we will use the following notations. For $W = (w_1, \ldots, w_{2p})$, we define $\widehat{y}_c(W) = \sum_{i=1}^{2p} D_i X w_i$. For $\theta = (U, \alpha)$, where $U \in \mathbb{R}^{d \times m}$ and $\alpha \in \mathbb{R}^m$, we define $\widehat{y}(\theta) = \sum_{j=1}^m \sigma(X u_j) \alpha_j$.

### B.1   PROOF OF PROPOSITION 1

According to the construction of $\mathcal{M}(\theta)$, there is at most one non-zero neuron per cone, and, each neuron $(w, \gamma)$ satisfies $\|w\| = |\gamma|$, i.e., $\mathcal{M}(\theta)$ is minimal.

Fix a cone $B$. Let $(w_1, \gamma_1), \ldots, (w_k, \gamma_k)$ be the neurons of $\theta$ such that $B(u_i, w_i) = B$ for $i = 1, \ldots, k$, and let $w^m = \frac{\sum_{j=1}^k |\gamma_j| w_j}{\sqrt{\sum_{j=1}^k |\gamma_j| w_j}}$ and $\gamma^m = \text{sign}(\gamma_1) \cdot \sqrt{\|w^m\|_2}$ be the merged neuron. Since $\theta$ is nearly minimal, we know that $w_1, \ldots, w_k$ are positively colinear.

For $t \in [0, 1]$, define $\theta(t)$ such that it has $k$ neurons associated with the cone $B$ given by

$$w_1(t) := \frac{(1-t)w_1|\gamma_1| + tw^m|\gamma^m|}{\sqrt{\|(1-t)w_1|\gamma_1| + tw^m|\gamma^m|\|_2}}$$
$$\gamma_1(t) := \text{sign}(\gamma_1) \cdot \|w_1(t)\|_2$$
$$w_j(t) := \sqrt{1-t} \cdot w_j$$
$$\gamma_j(t) := \sqrt{1-t} \cdot \gamma_j,$$

where $j \geqslant 2$. Note that for all $j \geqslant 1$, all vectors $w_j(t), w_j, w^m$ are positively colinear, that $\|w_j(t)\|_2 = |\gamma_j(t)|$ and that $\text{sign}(\gamma_j(t)) = \text{sign}(\gamma_1)$. Further, note that $(w_1(0), \gamma_1(0)) = (w_1, \gamma_1)$, $(w_1(1), \gamma_1(1)) = (w^m, \gamma^m)$ and for $j \geqslant 2$, $(w_j(0), \gamma_j(0)) = (w_j, \gamma_j)$, $(w_j(1), \gamma_j(1)) = (0, 0)$. Consider the neural networks $\theta(t)$ with neurons $(w_j(t), \gamma_j(t))$ respectively defined for each cone $B$. It holds that $\theta(0) = \theta$ and $\theta(1) = \mathcal{M}(\theta)$. Then, the contribution of the neurons in $B$ to the predictions $\widehat{y}(\theta(t))$ are given by

$$\sum_{j=1}^k \sigma(X w_j(t)) \gamma_j(t) = \text{sign}(\gamma) \cdot \sigma\left( X(w_1(t)|\gamma_1(t)| + \sum_{j=2}^k w_j(t)|\gamma_j(t)|) \right)$$
$$= \text{sign}(\gamma) \cdot \sigma\left( (1-t)X w_1|\gamma_1| + t X w^m|\gamma^m| + (1-t)X \sum_{j=2}^k w_j|\gamma_j| \right)$$
$$= \text{sign}(\gamma) \cdot \sigma(X w^m|\gamma^m|).$$

Thus, the predictions $\widehat{y}(\theta(t))$ are constant as a function of $t$. Similarly, we claim that the regularization term $R(\theta(t))$ is constant as a function of $t$. Since the neurons are scaled, the contribution of the cone $B$ to the regularization term is given by (up to the constant $\beta$)

$$
\begin{aligned}
\sum_{j=1}^{k} \|w_j(t)\|_2 |\gamma_j(t)| &= \|w_1(t)\|_2 |\gamma_1(t)| + \sum_{j=2}^{k} \|w_j(t)\|_2 |\gamma_j(t)| \\
&= \|(1-t)w_1|\gamma_1| + t\, w^m |\gamma^m|\|_2 + (1-t) \sum_{j=2}^{k} \|w_j\|_2 |\gamma_j| \\
&= \|(1-t) \sum_{j=1}^{k} w_j|\gamma_j| + t\, w^m |\gamma^m|\|_2 \\
&= \|w^m\|_2 |\gamma^m|,
\end{aligned}
$$

where the third equality follows from the triangular equality when all vectors are positively colinear. Thus, we have explicited a continuous path from $\theta$ to $\mathcal{M}(\theta)$ such that $\mathcal{L}_\beta$ is constant along that path.

Now, we show that if $\mathcal{M}(\theta)$ is a local minimum then $\theta$ is also a local minimum. We proceed with the converse. Assume that $\theta$ is not a local minimum of $\mathcal{L}_\beta$. For a cone $B$, let $(w_1, \gamma_1), \ldots, (w_k, \gamma_k)$ be the neurons of $\theta$ such that $B(w_i, \gamma_i) = B$. Let $(w^m, \gamma^m)$ be the merged neuron. If $\theta$ is not a local minimum, then there exists a small perturbation $\theta^\varepsilon := \{(u_i^\varepsilon, \alpha_i^\varepsilon)\}_{i=1}^{m}$ of the neurons of $\theta$ such that (i) $\mathcal{L}_\beta(\theta^\varepsilon) < \mathcal{L}_\beta(\theta)$, (ii) $\mathrm{sign}(\gamma_i^\varepsilon) = \mathrm{sign}(\gamma_i)$ and (iii) for each $i = 1, \ldots, m$, we have $I_+(\sigma(Xu_i)) \subseteq I_+(\sigma(Xu_i^\varepsilon))$ and $I_-(\sigma(Xu_i)) \subseteq I_-(\sigma(Xu_i^\varepsilon))$, where we use the notation $I_+(z) := \{i \in \{1, \ldots, n\} \mid z_i > 0\}$ and $I_-(z) := \{i \in \{1, \ldots, n\} \mid z_i < 0\}$ for a vector $z \in \mathbb{R}^n$.

Then, we define for $t \in [0, 1]$,

$$
\begin{aligned}
w_1(t) &= \frac{(1-t)w_1^\varepsilon |\gamma_1^\varepsilon| + t w^m |\gamma^m|}{\sqrt{\|(1-t)w_1^\varepsilon |\gamma_1^\varepsilon| + t w^m |\gamma^m|\|_2}}, \\
\gamma_1(t) &= \mathrm{sign}(\gamma_1) \cdot \|w_1(t)\|_2, \\
w_j(t) &= \sqrt{1-t} \cdot w_j^\varepsilon, \\
\gamma_j(t) &= \sqrt{1-t} \cdot \gamma_j^\varepsilon,
\end{aligned}
$$

where $j \geqslant 2$. Let $\theta(t)$ the neural network with neurons $(w_j(t), \gamma_j(t))$ defined as above for each cone $B$. Then, the contribution of a cone $B$ to the predictions $\widehat{y}(\theta(t))$ is given by

$$
\sum_{j=1}^{k} \sigma(Xw_j(t))\gamma_j(t) = \mathrm{sign}(\gamma_1)\left(\sigma(X((1-t)w_1^\varepsilon |\gamma_1^\varepsilon| + tw^m |\gamma^m|)) + (1-t)\sum_{j=2}^{k} \sigma(Xw_j^\varepsilon |\gamma_j^\varepsilon|)\right)
$$

Due to the above property (iii), we have that

$$
\sigma(X((1-t)w_1^\varepsilon |\gamma_1^\varepsilon| + tw^m |\gamma^m|)) = (1-t)\sigma(Xw_1^\varepsilon |\gamma_1^\varepsilon|) + t\,\sigma(Xw^m |\gamma^m|).
$$

Thus, the contribution of the cone $B$ to the predictions is

$$
\sum_{j=1}^{k} \sigma(Xw_j(t))\gamma_j(t) = (1-t)\sum_{j=1}^{k} \sigma(Xw_j^\varepsilon)\gamma_j^\varepsilon + t\,\sigma(Xw^m)\gamma^m.
$$

Summing over all the cones, we find that

$$
\widehat{y}(\theta(t)) = (1-t)\widehat{y}(\theta^\varepsilon) + t\,\widehat{y}(\mathcal{M}(\theta)).
$$

Similarly, the contribution of the cone $B$ to the regularization term $R(\theta(t))$ is

$$
\begin{aligned}
\sum_{j=1}^{k} \|w_j(t)\|_2 |\gamma_j(t)| &= \|(1-t)w_1^\varepsilon |\gamma_1^\varepsilon| + tw^m |\gamma^m|\|_2 + (1-t)\sum_{j=2}^{k} \|w_j^\varepsilon\|_2 |\gamma_j^\varepsilon| \\
&\leqslant (1-t)\sum_{j=1}^{k} \|w_j^\varepsilon\|_2 |\gamma_j^\varepsilon| + t\,\|w^m\|_2 |\gamma^m|,
\end{aligned}
$$

where the last inequality is due to the triangular inequality. Thus, we obtain that

$$R(\theta(t)) \leqslant (1-t)R(\theta^\varepsilon) + t\,R(\mathcal{M}(\theta))\,.$$

Hence, we get that $\mathcal{L}_\beta(\theta(t)) \leqslant (1-t)\mathcal{L}_\beta(\theta^\varepsilon) + t\,\mathcal{L}_\beta(\mathcal{M}(\theta))$. Since $\mathcal{L}_\beta(\theta^\varepsilon) < \mathcal{L}_\beta(\theta) = \mathcal{L}_\beta(\mathcal{M}(\theta))$, we have that $\mathcal{L}_\beta(\theta(t)) < \mathcal{L}_\beta(\mathcal{M}(\theta))$ for any $t < 1$. This concludes the proof.

### B.2 PROOF OF PROPOSITION 2

Let $\theta \in \Theta_m$, and consider the point $W(\theta)$, as defined in (7), whose expression is given by

$$w_i(\theta) := \sum_{\substack{j=1,\ldots,m \\ B(u_j,\alpha_j) \subseteq C_i}} |\alpha_j| u_j\,, \tag{17}$$

and such that each non-zero neuron $(u_j, \alpha_j)$ contributes only to a single $w_i(\theta)$.

We prove that the mapping $\theta \mapsto W(\theta)$ is well-defined. Each set $P_i$ is a cone and $w_i(\theta)$ is a positive linear combination of elements of $P_i$. It follows that $w_i(\theta) \in P_i$, and $W(\theta) \in \mathcal{W}$. Further, $\theta$ has $m$ neurons and each neuron $(u_j, \alpha_j)$ contributes only to a single $w_i(\theta)$. It follows that at most $m$ variables among $\{w_1(\theta), \ldots, w_{2p}(\theta)\}$ are non-zero, and $W(\theta) \in \mathcal{W}_m$. Hence, the mapping $\theta \mapsto W(\theta)$ is well-defined from $\Theta_m$ to $\mathcal{W}_m$.

We show that $\mathcal{L}_\beta^c(W(\theta)) \leqslant \mathcal{L}_\beta(\theta)$. We note that

$$\widehat{y}_c(W(\theta)) = \sum_{i=1}^{2p} D_i X w_i(\theta) = \sum_{i=1}^{2p} \sum_{\substack{j=1,\ldots,m \\ B(u_j,\alpha_j) \subseteq C_i}} D_i X |\alpha_j| u_j\,.$$

Note that for a neuron $(u_j, \alpha_j)$ such that $B(u_j, \alpha_j) \subseteq C_i$, we have that $D_i X |\alpha_j| u_j = \sigma(X u_j)\,\alpha_j$. It implies that

$$\widehat{y}_c(W(\theta)) = \sum_{i=1}^{2p} \sum_{\substack{j=1,\ldots,m \\ B(u_j,\alpha_j) \subseteq C_i}} D_i X |\alpha_j| u_j = \sum_{j=1}^m \sigma(X u_j)\alpha_j = \widehat{y}(\theta)\,,$$

and consequently, $\ell(\widehat{y}_c(W(\theta))) = \ell(\widehat{y}(\theta))$. On the other hand, we have

$$\sum_{i=1}^{2p} \|w_i\|_2 = \sum_{i=1}^{2p} \left\| \sum_{\substack{j=1,\ldots,m \\ B(u_j,\alpha_j) \subseteq C_i}} |\alpha_j| u_j \right\|_2 \leqslant \sum_{j=1}^m |\alpha_j| \|u_j\|_2\,,$$

where the last inequality follows from triangular inequality. Since the neurons $(u_j, \alpha_j)$ are scaled, we get that $\sum_{j=1}^m |\alpha_j| \|u_j\|_2 = \frac{1}{2} \sum_{j=1}^m |\alpha_j|^2 + \|u_j\|_2^2$, and we finally obtain that $\mathcal{L}_\beta^c(W(\theta)) \leqslant \mathcal{L}_\beta(\theta)$.

We show that the mapping $W \mapsto \theta(W)$ is well-defined. Based on the construction (8), it holds that the non-zero neurons $(u_j, \alpha_j)$ belong to pairwise distinct cones in $\{B_1, \ldots, B_{2q}\}$. Further, the neurons are scaled. Hence, $\theta(W)$ is a minimal neural network with $m$ neurons.

We prove that $\mathcal{L}_\beta(\theta(W)) \leqslant \mathcal{L}_\beta^c(W)$. Denote by $D^{(j)}$ the diagonal matrix associated with $B(u_j, \alpha_j)$ for $1 \leqslant j \leqslant l + l'$. We have that

$$\widehat{y}(\theta(W)) = \sum_{j=1}^m \sigma(X u_j)\alpha_j = \sum_{j=1}^{l+l'} D^{(j)} X u_j |\alpha_j| = \sum_{j=1}^{l+l'} D^{(j)} X \sum_{i \in K_j} v_i\,.$$

It holds by construction that for each $i \in K_j$, $D^{(j)} X w_i = D_i X w_i$. Therefore, we find that $\widehat{y}(\theta(W)) = \widehat{y}_c(W)$ and $\ell(\widehat{y}(\theta(W))) = \ell(\widehat{y}_c(W))$. On the other hand, we have that

$$\sum_{j=1}^m \|u_j\|_2 |\alpha_j| = \sum_{j=1}^{\ell+\ell'} \left\| \sum_{i \in K_j} v_i \right\|_2 \underset{(i)}{\leqslant} \sum_{j=1}^{\ell+\ell'} \sum_{i \in K_j} \|v_i\|_2 = \sum_{i=1}^{2p} \|v_i\|_2\,,$$

where inequality (i) follows from triangular inequality. Hence, we obtain that $\mathcal{L}_\beta(\theta(v)) \leqslant \mathcal{L}_\beta^c(v)$.

### B.3 Proof of Theorem 1

First, we show that $\tilde{\Theta}_m^{\mathrm{cvx}} \subseteq \Theta_m^*$. Let $\theta \in \Theta_m^{\mathrm{cvx}}$, and consider $\theta'$ a split version of $\theta$. Let $(u, \alpha)$ be a neuron of $\theta$, and $\{(u_j, \alpha_j)\}_{j=1}^k$ the neurons of $\theta'$ which correspond to the split of $(u, \alpha)$. We have $\sum_{j=1}^m \sigma(X u_j) \alpha_j = \sum_{j=1}^m \gamma_j \sigma(X u) \alpha = \sigma(X u) \alpha$ because $\sum_{j=1}^k \gamma_j = 1$. Furthermore, $\frac{1}{2} \sum_{j=1}^k \|u_j\|_2^2 + |\alpha_j|^2 = \frac{1}{2} \sum_{j=1}^k \gamma_j (\|u\|_2^2 + |\alpha|^2) = \frac{1}{2}(\|u\|_2^2 + |\alpha|^2)$, whence $\mathcal{L}(\theta) = \mathcal{L}(\theta')$. Consequently, $\tilde{\Theta}_m^{\mathrm{cvx}} \subseteq \Theta_m^*$.

It remains to show that $\Theta_m^* \subseteq \tilde{\Theta}_m^{\mathrm{cvx}}$. Let $\theta \in \Theta_m^*$. Due to the strong convexity of the regularization term and the re-scaling invariance of the term $\sigma(X u_j) \alpha_j$, we must have that $\|u_j\|_2 = |\alpha_j|$ for each neuron $(u_j, \alpha_j)$ of $\theta$. We partition the neurons of $\theta$ such that the neurons in each partition $\{(u_i, \alpha_i)\}_{i=1}^k$ belong to the same cone $C$ (in the sense that $u_i \alpha_i \in C$), and the cones are pairwise distinct across partitions. Due to the regularization term, it is straightforward to show that the neurons must be positively colinear, and that this corresponds to a split. Thus, $\Theta_m \subseteq \tilde{\Theta}_m^{\mathrm{cvx}}$ and this concludes the proof.

### B.4 Proof of Theorem 2: Clarke stationary points are nearly minimal neural networks

We review the definition of the *Clarke subdifferential* Clarke (1975) of $f$. At $x \in \mathbb{R}^d$, this is defined as

$$\partial_C f(x) := \mathrm{conv} \left\{ \lim_{k \to \infty} \nabla f(x_k) \mid \lim_{k \to \infty} x_k = x, \ x_k \in D \right\},$$

where $D := \{x \in \mathbb{R}^d \mid f \text{ differentiable at } x\}$. In particular, it holds that $\mathbb{R}^d \setminus D$ has measure equal to zero Borwein & Lewis (2010) under mild assumptions on $f$. Then, we say that $x \in \mathbb{R}^d$ is *Clarke stationary* with respect to $f$ if $0 \in \partial_C f(x)$.

Let $\theta \in \Theta_m$ be Clarke stationary, i.e., there exist $\lambda^{(1)}, \ldots, \lambda^{(N)} > 0$ and sequences $\{\theta_k^{(1)}\}_k, \ldots, \{\theta_k^{(N)}\}_k$ such that $\sum_{j=1}^N \lambda^{(j)} = 1$, $\lim_{k \to \infty} \theta_k^{(j)} = \theta$ for each $j = 1, \ldots, N$, the loss function $\mathcal{L}_\beta$ is differentiable at each $\theta_k^{(j)}$ and

$$0 = \sum_{j=1}^N \lambda^{(j)} \lim_{k \to \infty} \nabla \mathcal{L}_\beta(\theta_k^{(j)}).$$

PART 1: THE NEURAL NETWORK $\theta$ MUST BE SCALED.

By contradiction, we assume first that the neural network $\theta$ is unscaled, i.e., there exists a neuron $(u, \alpha)$ such that $\|u\|_2 \neq |\alpha|$. Write $(u_k^{(j)}, \alpha_k^{(j)})$ the corresponding neuron of each $\theta_k^{(j)}$. Since $\lim_{k \to \infty} (u_k^{(j)}, \alpha_k^{(j)}) = (u, \alpha)$, up to extracting subsequences, we can assume that the neurons $(u_k^{(j)}, \alpha_k^{(j)})$ are also unscaled, i.e., $\|u_k^{(j)}\|_2 \neq |\alpha_k^{(j)}|$ for all $k \geqslant 1$ and $j = 1, \ldots, N$.

CASE 1: $\alpha \neq 0$

Since $\lim_{k \to \infty} \alpha_k^{(j)} = \alpha$, up to extracting subsequences, we can assume that $\alpha_k^{(j)} \neq 0$ for all $k \geqslant 1$ and $j = 1, \ldots, N$. Then, for each $j = 1, \ldots, N$ and $k \geqslant 1$ and for $t \in [0, 1]$, we define the neural network $\theta_k^{(j)}(t)$ as a copy of $\theta_k^{(j)}$ except for the neuron $(u_k^{(j)}, \alpha_k^{(j)})$ that we replace by

$$u_k^{(j)}(t) = \frac{u_k^{(j)}}{\gamma_k^{(j)}(t)}, \qquad \alpha_k^{(j)}(t) = \gamma_k^{(j)}(t) \cdot \alpha_k^{(j)},$$

where $\gamma_k^{(j)*} = \sqrt{\frac{\|u_k^{(j)}\|_2}{|\alpha_k^{(j)}|}}$, $\gamma_k^{(j)}(t) = 1 + t(\gamma_k^{(j)*} - 1)$ and we use the improper notation $\frac{u_k^{(j)}}{\gamma_k^{(j)}(t)} = 0$ if $u_k^{(j)} = 0$. Note that $\theta_k^{(j)}(t)$ defines a continuous path from $\theta_k^{(j)} = \theta_k^{(j)}(0)$ to the scaled neural network $\theta_k^{(j)}(1)$. Further, since $\sigma$ is positively homogeneous, it holds that for any $t \in [0, 1]$,

$$\sigma(X u_k^{(j)}(t)) \alpha_k^{(j)}(t) = \sigma(X u_k^{(j)}) \alpha_k^{(j)},$$

so that that the function $\mathcal{L}_\beta(\theta_k^{(j)}(t))$ is constant as a function of $t \in [0,1]$. On the other hand, the regularization term satisfies

$$R(\theta_k^{(j)}(t)) = \underbrace{R(\theta_k^{(j)}) - \frac{\beta}{2}(\|u_k^{(j)}\|_2^2 + |\alpha_k^{(j)}|^2)}_{:=C(k,j)} + \frac{\beta}{2}\left(\frac{\|u_k^{(j)}\|_2^2}{\gamma_k^{(j)2}(t)} + \gamma_k^{(j)2}(t)|\alpha_k^{(j)}|^2\right)$$

$$= \underbrace{C(k,j)}_{\text{independent of } t} + \frac{\beta}{2}\left(\frac{\|u_k^{(j)}\|_2^2}{\gamma_k^{(j)2}(t)} + \gamma_k^{(j)2}(t)|\alpha_k^{(j)}|^2\right).$$

Note that the function $g_k^{(j)}(t) := \mathcal{L}_\beta(\theta_k^{(j)}(t))$ is differentiable, and simple algebra yields

$$\frac{dg_k^{(j)}}{dt}(0) = \beta \cdot \left(\frac{\sqrt{\|u_k^{(j)}\|_2} - \sqrt{|\alpha_k^{(j)}|}}{\sqrt{|\alpha_k^{(j)}|}}\right) \cdot (|\alpha_k^{(j)}|^2 - \|u_k^{(j)}\|_2^2).$$

Hence,

$$\lim_{k\to\infty} \frac{dg_k^{(j)}}{dt}(0) = \beta \cdot \left(\frac{\sqrt{\|u\|_2} - \sqrt{|\alpha|}}{\sqrt{|\alpha|}}\right) \cdot (|\alpha|^2 - \|u\|_2^2).$$

Since $|\alpha| \neq \|u\|_2$, it follows that $\lim_{k\infty} \frac{dg_k^{(j)}}{dt}(0) < 0$. On the other hand, we have that

$$\frac{dg_k^{(j)}}{dt}(0) = \langle \frac{d\theta_k^{(j)}}{dt}(0), \nabla\mathcal{L}_\beta(\theta_k^{(j)}) \rangle.$$

Simple algebra yields that $\lim_{k\infty} \frac{du_k^{(j)}(t=0)}{dt} = \left(1 - \sqrt{\frac{\|u\|_2}{|\alpha|}}\right) \cdot u$ and $\lim_{k\infty} \frac{d\alpha_k^{(j)}(t=0)}{dt} = \left(\sqrt{\frac{\|u\|_2}{|\alpha|}} - 1\right) \cdot \alpha$. Thus, the limit $d\theta := \lim_{k\infty} \frac{d\theta_k^{(j)}}{dt}(0)$ is constant (independent of the index $j$) and

$$\sum_{j=1}^{N} \lambda^{(j)} \lim_{k\to\infty} \frac{dg_k^{(j)}}{dt}(0) = \left\langle d\theta, \underbrace{\sum_{j=1}^{N} \lambda^{(j)} \lim_{k\infty} \nabla\mathcal{L}_\beta(\theta_k^{(j)})}_{=0}, \right\rangle$$

$$= 0.$$

This is contradiction with the fact that $\sum_{j=1}^{N} \lambda^{(j)} \lim_{k\to\infty} \frac{dg_k^{(j)}}{dt}(0) < 0$. Therefore, in the case $u \neq 0$ and $\alpha \neq 0$, we must have that $\|u\|_2 = |\alpha|$.

CASE 2: $u \neq 0$

The proof proceeds exactly in the same way, except that we define

$$u_k^{(j)}(t) = \gamma_k^{(j)}(t) \cdot u_k^{(j)}, \qquad \alpha_k^{(j)}(t) = \frac{\alpha_k^{(j)}}{\gamma_k^{(j)}(t)},$$

where $\gamma_k^{(j)*} = \sqrt{\frac{|\alpha_k^{(j)}|}{\|u_k^{(j)}\|_2}}$, $\gamma_k^{(j)}(t) = 1 + t(\gamma_k^{(j)*} - 1)$ and we use the convention $\frac{\alpha_k^{(j)}}{\gamma_k^{(j)}(t)} = 0$ if $\alpha_k^{(j)} = 0$.

PART 2: NON-ZERO NEURONS WHICH SHARE THE SAME ACTIVATION CONE ARE POSITIVELY COLINEAR

According to the first part of the proof, we can assume that the neural network $\theta$ is scaled.

(SPECIAL CASE) THE NEURAL NETWORK $\theta$ IS A DIFFERENTIABLE POINT OF $\mathcal{L}_\beta$.

In order to provide some intuition about the proof, let us assume first that $\mathcal{L}_\beta$ is differentiable at $\theta$.

By contradiction, we suppose that there exist two non-zero neurons $(u, \alpha)$ and $(v, \beta)$ such that $B(u, \alpha) = B(v, \beta)$, and, $u$ and $v$ are not positively colinear. Further, let us assume that $\alpha, \beta > 0$ (the case $\alpha, \beta < 0$ follows the same lines).

Define $w := \alpha u + \beta v$. Note that $w$ has the same sign pattern as $u$ and $v$. For $t \in [0, 1]$, we set

$$\widetilde{u}(t) := (1 - t)\alpha u + \frac{t}{2}w \,,$$

$$\widetilde{v}(t) := (1 - t)\beta v + \frac{t}{2}w \,,$$

$$u(t) := \frac{\widetilde{u}(t)}{\sqrt{\|\widetilde{u}(t)\|_2}} \,, \qquad \alpha(t) := \sqrt{\|\widetilde{u}(t)\|_2} \,,$$

$$v(t) := \frac{\widetilde{v}(t)}{\sqrt{\|\widetilde{v}(t)\|_2}} \,, \qquad \beta(t) := \sqrt{\|\widetilde{v}(t)\|_2} \,.$$

Note that $B(u(t), \alpha(t)) = B(u, \alpha) = B(v, \beta) = B(v(t), \beta(t))$. Further, we define $\theta(t)$ as a copy of $\theta$ where we replace the two neurons $(u, \alpha)$ and $(v, \beta)$ by $(u(t), \alpha(t))$ and $(v(t), \beta(t))$. Note that $\theta(t)$ defines a continuous path in $\Theta_m$ starting at $\theta$.

Then, we introduce the two functions

$$g(t) := \ell(\widehat{y}(\theta(t))) \,,$$

$$h(t) := R(\theta(t)) \,,$$

so that $\mathcal{L}_\beta(\theta(t)) = g(t) + \beta \cdot h(t)$. First, we claim that $g(t)$ is constant over $[0, 1]$. Indeed, this follows from the fact

$$\sigma(Xu(t))\alpha(t) + \sigma(Xv(t))\beta(t) = \sigma(X\underbrace{(\widetilde{u}(t) + \widetilde{v}(t))}_{= \alpha u + \beta v}) = \sigma(Xu)\alpha + \sigma(Xv)\beta \,,$$

where the first equality comes from the fact that $B(u(t), \alpha(t)) = B(v(t), \beta(t))$. Hence, we have $\widehat{y}(\theta(t)) = \widehat{y}(\theta)$ and $g(t)$ is constant.

On the other hand, the function $h(t)$ is clearly differentiable, and simple algebra yields that

$$\frac{\mathrm{d}h(0)}{\mathrm{d}t} = -\frac{\|\alpha u\|_2}{2} - \frac{\|\beta v\|_2}{2} + \frac{1}{2}(\alpha u)^\top (\beta v) \left( \frac{1}{\|\alpha u\|_2} + \frac{1}{\|\beta v\|_2} \right) \,.$$

Since $u$ and $v$ are not colinear, it holds by Cauchy-Schwarz inequality that $(\alpha u)^\top (\beta v) < \|\alpha u\|_2 \|\beta v\|_2$, and thus,

$$\frac{\mathrm{d}h(0)}{\mathrm{d}t} < -\frac{\|\alpha u\|_2}{2} - \frac{\|\beta v\|_2}{2} + \frac{\|\alpha u\|_2 \|\beta v\|_2}{2} \left( \frac{1}{\|\alpha u\|_2} + \frac{1}{\|\beta v\|_2} \right) = 0 \,,$$

that is, $\frac{\mathrm{d}h(0)}{\mathrm{d}t} < 0$. Thus, we finally obtain that $\frac{\mathrm{d}\mathcal{L}_\beta(\theta(0))}{\mathrm{d}t} < 0$, which contradicts the stationarity of $\theta$.

(GENERAL CASE) THE NEURAL NETWORK $\theta$ IS NOT NECESSARILY A DIFFERENTIABLE POINT OF $\mathcal{L}_\beta$.

Now, let us generalize the above proof to the case where $\mathcal{L}_\beta$ is not necessarily differentiable at $\theta$.

For a vector $z \in \mathbb{R}^n$, we use the notations $I_+(z) := \{i \in \{1, \ldots, n\} \mid z_i > 0\}$, $I_0(z) := \{i \in \{1, \ldots, n\} \mid z_i = 0\}$ and $I_-(z) := \{i \in \{1, \ldots, n\} \mid z_i < 0\}$.

Since $\theta$ is a Clarke stationary point of $\mathcal{L}_\beta$, we know that there exist $\lambda^{(1)}, \ldots, \lambda^{(N)} > 0$ and sequences $\{\theta_k^{(1)}\}_k, \ldots, \{\theta_k^{(N)}\}_k$ such that $\sum_{j=1}^N \lambda^{(j)} = 1$, $\lim_{k \to \infty} \theta_k^{(j)} = \theta$ for each $j = 1, \ldots, N$, the loss function $\mathcal{L}_\beta$ is differentiable at each $\theta_k^{(j)}$ and

$$0 = \sum_{j=1}^N \lambda^{(j)} \lim_{k \to \infty} \nabla \mathcal{L}_\beta(\theta_k^{(j)}) \,.$$

For each $k \geqslant 1$ and $j = 1, \ldots, N$, up to extracting subsequences, we can assume that $u_k^{(j)}, \alpha_k^{(j)}, v_k^{(j)}, \beta_k^{(j)} \neq 0$, and, $\alpha_k^{(j)}$ and $\beta_k^{(j)}$ have the same sign (let us say positive). Further, up to extracting subsequences again, we can assume that the sign patterns $I_+(Xu_k^{(j)})$ and $I_-(Xu_k^{(j)})$ (resp. $I_+(Xv_k^{(j)})$ and $I_-(Xv_k^{(j)})$) remain constant (independent of $k$). Since the sign patterns of $Xu$ and $Xv$ are equal by assumption, and, since $\lim_{k\infty} u_k^{(j)} = u$ and $\lim_{k\infty} v_k^{(j)} = v$, it follows that

$$I_+(Xu) = I_+(Xv) \subset \{I_+(Xu_k^{(j)}) \cap I_+(Xv_k^{(j)})\}, \tag{18}$$

and

$$I_-(Xu) = I_-(Xv) \subset \{I_-(Xu_k^{(j)}) \cap I_-(Xv_k^{(j)})\}. \tag{19}$$

We denote $T^{(j)}(u)$ and $T^{(j)}(v)$ the diagonal matrices (as introduced in Section 2) which correspond to the sign patterns of $u_k^{(j)}$ and $v_k^{(j)}$, and which are independent of $k$ by assumption. Then, using (18) and (19), it follows that

$$T^{(j)}(u)Xu = T^{(j)}(v)Xv, \qquad T^{(j)}(u)Xu = T^{(j)}(v)Xv. \tag{20}$$

The above equalities will be crucial later on in our analysis.

Then, for each neural network $\theta_k^{(j)}$, we can construct a similar path $\theta_k^{(j)}(t)$ as in the differentiable case, that is, we set $w_k^{(j)} := \alpha_k^{(j)} u_k^{(j)} + \beta_k^{(j)} v_k^{(j)}$, and

$$\widetilde{u}_k^{(j)}(t) := (1-t)\alpha u_k^{(j)} + \frac{t}{2} w_k^{(j)},$$

$$\widetilde{v}_k^{(j)}(t) := (1-t)\beta_k^{(j)} v_k^{(j)} + \frac{t}{2} w_k^{(j)},$$

$$u_k^{(j)}(t) := \frac{\widetilde{u}_k^{(j)}(t)}{\sqrt{\|\widetilde{u}_k^{(j)}(t)\|_2}}, \qquad \alpha_k^{(j)}(t) := \sqrt{\|\widetilde{u}_k^{(j)}(t)\|_2},$$

$$v_k^{(j)}(t) := \frac{\widetilde{v}_k^{(j)}(t)}{\sqrt{\|\widetilde{v}_k^{(j)}(t)\|_2}}, \qquad \beta_k^{(j)}(t) := \sqrt{\|\widetilde{v}_k^{(j)}(t)\|_2}.$$

Similarly to the differentiable case, we also define the functions

$$g_k^{(j)}(t) := \ell(\widehat{y}(\theta_k^{(j)}(t))),$$

$$h_k^{(j)}(t) := R(\theta_k^{(j)}(t)).$$

First, we claim that $\lim_{k\to\infty} \frac{\mathrm{d}g_k^{(j)}(0)}{\mathrm{d}t} = 0$. Indeed, we have

$$\frac{\mathrm{d}g_k^{(j)}(0)}{\mathrm{d}t} = \frac{1}{2} \left\langle (T^{(j)}(u) - T^{(j)}(v))X(v_k^{(j)} - u_k^{(j)}), \nabla\ell(\widehat{y}(\theta_k^{(j)})) \right\rangle.$$

Taking the limit $k \to \infty$, we obtain that

$$\lim_{k\to\infty} \frac{\mathrm{d}g_k^{(j)}(0)}{\mathrm{d}t} = \frac{1}{2} \left\langle (T^{(j)}(u) - T^{(j)}(v))X(v - u), \nabla\ell(\widehat{y}(\theta)) \right\rangle.$$

Using (20), we get that $(T^{(j)}(u) - T^{(j)}(v))X(v - u) = 0$, and consequently, the claimed equality $\lim_{k\to\infty} \frac{\mathrm{d}g_k^{(j)}(0)}{\mathrm{d}t} = 0$.

On the other hand, the function $h_k^{(j)}(t)$ is clearly differentiable, and simple algebra yields that

$$\lim_{k\to\infty} \frac{\mathrm{d}h_k^{(j)}(0)}{\mathrm{d}t} = -\frac{\|\alpha u\|_2}{2} - \frac{\|\beta v\|_2}{2} + \frac{1}{2}(\alpha u)^\top(\beta v)\left(\frac{1}{\|\alpha u\|_2} + \frac{1}{\|\beta v\|_2}\right).$$

Since $u$ and $v$ are not colinear, it holds by Cauchy-Schwarz inequality that $(\alpha u)^\top(\beta v) < \|\alpha u\|_2\|\beta v\|_2$, and thus,

$$\lim_{k\to\infty} \frac{\mathrm{d}h_k^{(j)}(0)}{\mathrm{d}t} < -\frac{\|\alpha u\|_2}{2} - \frac{\|\beta v\|_2}{2} + \frac{\|\alpha u\|_2\|\beta v\|_2}{2}\left(\frac{1}{\|\alpha u\|_2} + \frac{1}{\|\beta v\|_2}\right) = 0,$$

that is, $\lim_{k\to\infty} \frac{\mathrm{d}h_k^{(j)}(0)}{\mathrm{d}t} < 0$. Thus, we finally obtain that

$$\lim_{k\to\infty} \frac{\mathrm{d}\mathcal{L}_\beta(\theta_k^{(j)}(0))}{\mathrm{d}t} < 0\,,$$

and further, that

$$\sum_{j=1}^{N} \lambda^{(j)} \lim_{k\to\infty} \frac{\mathrm{d}\mathcal{L}_\beta(\theta_k^{(j)}(0))}{\mathrm{d}t} < 0\,.$$

However, it holds that

$$\lim_{k\to\infty} \frac{\mathrm{d}\mathcal{L}_\beta(\theta_k^{(j)}(0))}{\mathrm{d}t} = \lim_{k\to\infty} \langle \frac{\mathrm{d}\theta_k^{(j)}(0)}{\mathrm{d}t}, \nabla\mathcal{L}_\beta(\theta_k^{(j)}) \rangle$$

It is immediate to see that $\mathrm{d}\theta := \lim_{k\infty} \frac{\mathrm{d}\theta_k^{(j)}(0)}{\mathrm{d}t}$ does not depend on the index $j$, so that

$$\sum_{j=1}^{N} \lambda^{(j)} \lim_{k\to\infty} \frac{\mathrm{d}\mathcal{L}_\beta(\theta_k^{(j)}(0))}{\mathrm{d}t} = \langle \mathrm{d}\theta, \underbrace{\sum_{j=1}^{N} \lambda^{(j)} \lim_{k\to\infty} \nabla\mathcal{L}_\beta(\theta_k^{(j)})}_{=\,0} \rangle\,.$$

That is, we obtained both that $\sum_{j=1}^{N} \lambda^{(j)} \lim_{k\to\infty} \frac{\mathrm{d}\mathcal{L}_\beta(\theta_k^{(j)}(0))}{\mathrm{d}t} < 0$ and $\sum_{j=1}^{N} \lambda^{(j)} \lim_{k\to\infty} \frac{\mathrm{d}\mathcal{L}_\beta(\theta_k^{(j)}(0))}{\mathrm{d}t} = 0$, which is a contradiction. This concludes the proof that $\theta$ must be a nearly minimal neural network.

## B.5 PROOF OF PROPOSITION 3: REDUCTION TO NEARLY MINIMAL NEURAL NETWORKS ALONG A PATH OF DECREASING OBJECTIVE VALUE

We consider reductions similar to those in the proof of Theorem 2, in order to construct a path $\theta(t) \in \Theta_m$ for $t \in [0,1]$ such that $\theta(0) = \theta$, $\theta(1) \in \widetilde{\Theta}_m^{\min}$ and $\mathcal{L}_\beta(\theta(t))$ is strictly decreasing. Naturally, we assume that $\theta$ is not nearly minimal, otherwise, there is nothing to show.

PART 1: THE NEURAL NETWORK $\theta$ IS UNSCALED.

We claim that there exists a path $\theta(t) \in \Theta_m$ for $t \in [0,1]$ such that $\theta(0) = \theta$, $\theta(1)$ is scaled and $\mathcal{L}_\beta(\theta(t))$ is strictly decreasing.

Suppose that the neural network is unscaled (if not, go directly to Part 2). Then, for each neuron $(u,\alpha)$ of $\theta$ such that $\|u\|_2 \neq |\alpha|$, define

$$u(t) = \begin{cases} \sqrt{|\alpha|} \cdot \frac{u}{\gamma(t)} & \text{if } u, \alpha \neq 0, \\ 0 & \text{otherwise}, \end{cases}$$

$$\alpha(t) = \begin{cases} \gamma(t) \cdot \frac{\alpha}{\sqrt{|\alpha|}} & \text{if } \alpha \neq 0, \\ 0 & \text{otherwise}, \end{cases}$$

where $\gamma(t) = \sqrt{|\alpha|} + t(\sqrt{\|u\|_2} - \sqrt{|\alpha|})$. Simple algebra yields that $(u(0), \alpha(0)) = (u, \alpha)$ and $\|u(1)\|_2 = \sqrt{|\alpha| \|u\|_2} = |\alpha(1)|$, so that $\theta(0) = \theta$ and $\theta(1)$ is scaled. By positive homogeneity of $\sigma$, we have that $\sigma(Xu(t))\alpha(t) = \sigma(Xu)\alpha$, which further implies that $\widehat{y}(\theta(t)) = \widehat{y}(\theta)$ and $\mathcal{L}(\widehat{y}(\theta(t)))$ is constant as a function of $t$.

We claim that the regularization term $R(\theta(t))$ is strictly decreasing as a function of $t$. Indeed, it holds that

$$\|u(t)\|_2^2 + |\alpha(t)|^2 = \frac{|\alpha|}{\gamma^2(t)} \|u\|_2^2 + \frac{\gamma^2(t)}{|\alpha|} |\alpha|^2\,.$$

The minimizer of the function $\gamma \in \mathbb{R} \longmapsto \frac{|\alpha|}{\gamma^2} \|u\|_2^2 + \frac{\gamma^2}{|\alpha|} |\alpha|^2$ is given by $\gamma^* = \sqrt{\|u\|_2}$, which is also equal to $\gamma(1)$, and the minimal value of the latter function is given by $2\|u\|_2 |\alpha|$, which is strictly smaller than $\|u\|_2^2 + |\alpha|^2$ since $\|u\|_2 \neq |\alpha|$. Thus, the function $t \mapsto R(\theta(t))$ is minimized at $t = 1$, and $R(\theta(1)) < R(\theta)$. Lastly, observe that $t \mapsto R(\theta(t))$ is a convex function, which implies that it must be strictly decreasing over $[0,1]$. This concludes the first part of the proof.

PART 2: THE NEURAL NETWORK $\theta$ IS SCALED BUT NOT NEARLY MINIMAL.

If the neural network $\theta$ is scaled but not nearly minimal, we claim that there exists a continuous path $\theta(t)$ for $t \in [0, 1]$ such that $\theta(0) = \theta$, $\theta(1)$ is nearly minimal, and $\mathcal{L}_\beta(\theta(t))$ is strictly decreasing.

For each cone $B \in \{B_1, \dots, B_{2q}\}$, we consider the non-zero neurons $\mathcal{U}_B := \{(u, \alpha)\}$ of $\theta$ such that $B(u, \alpha) = B$. By assumption, there exists at least one cone $B$ such that $\mathcal{U}_B$ has more than two elements which are not positively colinear. Then, for each cone $B$, we set $w = \sum_{(u,\alpha)\in\mathcal{U}_B} |\alpha|u$, and, for each $(u, \alpha) \in \mathcal{U}_B$ and for $t \in [0, 1]$,

$$\widetilde{u}(t) := (1-t)|\alpha|u + \frac{t}{|\mathcal{U}_B|}w,$$

$$u(t) := \text{sign}(\alpha) \cdot \frac{\widetilde{u}(t)}{\sqrt{\|\widetilde{u}(t)\|_2}},$$

$$\alpha(t) := \text{sign}(\alpha) \cdot \sqrt{\|\widetilde{u}(t)\|_2},$$

where $|\mathcal{U}_B|$ is the cardinality of the set $\mathcal{U}_B$. Note that $B(u(t), \alpha(t)) = B(u, \alpha) = B$, and each neuron $(u(t), \alpha(t))$ is scaled. Further, we define $\theta(t)$ the neural network with neurons $(u(t), \alpha(t))$. It holds that $\theta(t)$ defines a continuous path in $\Theta_m$ starting at $\theta$, and ending at a nearly minimal neural network. Then, we introduce the two function $g(t) = \mathcal{L}(\widehat{y}(\theta(t)))$ and $h(t) := R(\theta(t))$, so that $\mathcal{L}_\beta(\theta(t)) = g(t) + \beta \cdot h(t)$. First, we claim that $g(t)$ is constant over $[0, 1]$. Indeed, this comes from the fact that for each cone $B$,

$$\sum_{(u,\alpha)\in\mathcal{U}_B} \sigma(Xu(t))\alpha(t) = \text{sign}(\alpha) \cdot \sigma(X \cdot \underbrace{\sum_{(u,\alpha)\in\mathcal{U}_B} |\alpha(t)|u(t)}_{=w})$$

$$= \text{sign}(\alpha) \cdot \sigma(Xw)$$

$$= \sum_{(u,\alpha)\in\mathcal{U}_B} \sigma(Xu)\alpha.$$

The first (resp. third) equality holds from the fact that the neurons $(u(t), \alpha(t))$ (resp. $(u, \alpha)$) have the same active cone $B$. Thus, $\widehat{y}(\theta(t)) = \widehat{y}(\theta)$ and $g(t)$ is indeed constant.

On the other hand, we claim that the function $h(t)$ is strictly decreasing. Indeed, observe first that

$$h(t) = \frac{\beta}{2} \sum_B \sum_{(u,\alpha)\in\mathcal{U}_B} \|u(t)\|_2^2 + |\alpha(t)|^2$$

$$= \beta \sum_B \sum_{(u,\alpha)\in\mathcal{U}_B} \|u(t)\|_2|\alpha(t)|$$

$$= \beta \sum_B \sum_{(u,\alpha)\in\mathcal{U}_B} \|\widetilde{u}(t)\|_2,$$

where the second equality holds since the neurons $(u(t), \alpha(t))$ are scaled. Thus, it is immediate to verify that the function $h$ is differentiable, and

$$h'(t) = \beta \cdot \sum_B \sum_{(u,\alpha)\in\mathcal{U}_B} \frac{1}{\|\widetilde{u}(t)\|_2} \left( t \cdot \|\frac{w}{|\mathcal{U}_B|} - |\alpha|u\|_2^2 + |\alpha|u^\top(\frac{w}{|\mathcal{U}_B|} - |\alpha|u) \right).$$

Clearly, $h'(t)$ is strictly increasing (since there exists, by assumption, at least one cone $B$ and a neuron $(u, \alpha) \in \mathcal{U}_B$ such that $\frac{w}{|\mathcal{U}_B|} \neq |\alpha|u$). Therefore, it suffices to verify that $h'(1) \leqslant 0$. Simple algebra yields actually that $h'(1) = 0$, which concludes the proof.

B.6   PROOF OF PROPOSITION 4

The proof with trichotomies is almost identical to the proof of dichotomies in Pilanci & Ergen (2020); Sahiner et al. (2020). We start with the dual representation of $\mathcal{P}^*$:

$$\mathcal{P}^* = \max \ell^*(\lambda), \text{ s.t. } \max_{w:\|w\|_2 \leqslant 1} |\lambda^T(Xw)_+| \leqslant \beta.$$

Here $\ell^*(\lambda) := \max_v\{\lambda^T v - \ell(v)\}$ is the Fenchel conjugate function of $\ell$. We note that the single-sided dual constraint has an equivalent formulation using trichotomies:

$$\max_{w:\|w\|_2 \leqslant 1} \lambda^T(Xw)_+$$
$$= \max_{j\in[q]} \max_{w:\|w\|_2 \leqslant 1, w\in Q_j} \lambda^T(T_j)_+Xw.$$

Similarly, the other side of the dual constraint can be formulated as

$$\max_{w:\|w\|_2 \leqslant 1} -\lambda^T(Xw)_+$$
$$= \max_{i\in[q]} \max_{w:\|w\|_2 \leqslant 1, w\in Q_i} \lambda^T(T_j)_+X(-w).$$

Therefore, we can rewrite $\mathcal{P}^*$ as

$$\mathcal{P}^* = \max \ell^*(\lambda),$$
$$\text{s.t.} \max_{w:\|w\|_2 \leqslant 1, w\in Q_i} \lambda^T(T_j)_+Xw \leqslant \beta, i\in[q],$$
$$\max_{w:\|w\|_2 \leqslant 1, w\in Q_i} \lambda^T(T_j)_+X(-w) \leqslant \beta, j\in[q].$$

For simplicity, we denote $T_{j+q} = T_j$ for $j\in[q]$. We now formulate the Lagrangian

$$\mathcal{P}^* = \max_\lambda \min_{\nu \geqslant 0} \min_{\substack{w_j\in Q_j, \|w_j\|_2 \leqslant 1 \\ w_{j+q}\in Q_j, \|w_{j+q}\|_2 \leqslant 1}} \ell^*(\lambda) + \sum_{j=1}^q \nu_j(\beta - \lambda^T(T_j)_+Xw_j)$$
$$+ \sum_{j=1}^q \nu_{j+q}(\beta - \lambda^T(T_j)_+X(-w_{j+q})).$$

By Sion's minimax theorem, we can switch the max and min, and then minimize over $\lambda$. Following this, we obtain

$$\mathcal{P}^* = \min_{\nu_j \geqslant 0} \min_{\substack{w_j\in Q_j, \|w_j\|_2 \leqslant 1 \\ w_{j+q}\in Q_i, \|w_{j+q}\|_2 \leqslant 1}} \ell\left(\sum_{j=1}^q (T_i)_+X(\nu_i w_i - \nu_{j+q}w_{j+q})\right) + \beta\sum_{j=1}^{2q} \nu_j.$$

By rescaling the variable $w_j = \nu_j w_j$, we can reformulate $P^*$ as

$$\mathcal{P}^* = \min_{\nu_i \geqslant 0} \min_{\substack{w_j\in Q_j, \|w_j\|_2 \leqslant \nu_j \\ w_{j+q}\in Q_j, \|w_{j+q}\|_2 \leqslant \nu_{j+q}}} \ell\left(\sum_{j=1}^q (T_j)_+X(w_j - w_{j+q})\right) + \beta\sum_{j=1}^{2q} \nu_j.$$

Minimizing with respect to $\nu$ yields

$$\mathcal{P}^* = \min_{w_j, w_{j+q}\in Q_j} \ell\left(\sum_{j=1}^q (T_j)_+X(\nu_j w_j - \nu_{j+q}w_{j+q})\right) + \beta\sum_{j=1}^{2q} \|w_j\|_2.$$

This completes the proof.

### B.7 Proof of Theorem 3

According to Proposition 1 and 2, we can assume that $\theta$ is a minimal neural network. Denote $\tilde{\lambda} = \nabla\ell\left(\sum_{j=1}^m (Xu_j)_+\alpha_j\right)$. From the definition of Clarke's stationary point, for $j\in[m]$ with $u_j \neq 0$, we have

$$-\beta u_j \in \partial_{u_j}^\circ \ell\left(\sum_{j=1}^m (Xu_j)_+\alpha_j\right),$$
$$-\beta\alpha_j = \tilde{\lambda}^T(Xu_j)_+.$$

(21)

The first line in (21) is equivalent to that there exists $\delta_j \in [0, 1]^N$ such that

$$-\beta u_j = \alpha_j(X^T \tilde{D}_j \tilde{\lambda} + X^T \tilde{S}_j \text{diag}(\delta_j) \tilde{\lambda}). \tag{22}$$

Here $\tilde{D}_j = \text{diag}(\mathbb{I}(Xu_j \geqslant 0))$ and $\tilde{S}_j = \text{diag}(\mathbb{I}(Xu_j = 0))$. As $u_j \neq 0$ and $\alpha_j \neq 0$, this implies that

$$-\beta \frac{u_j}{\alpha_j} = X^T \tilde{D}_j \tilde{\lambda} + X^T \tilde{S}_j \text{diag}(\delta_j) \tilde{\lambda}. \tag{23}$$

For the second line in (21), we can also rewrite it as

$$\begin{aligned}
-\beta \alpha_j &= \tilde{\lambda}^T \tilde{D}_j X u_j \\
&= u_j^T X^T \tilde{D}_j \tilde{\lambda} \\
&= u_j^T (X^T \tilde{D}_j \tilde{\lambda} + X^T \tilde{S}_j \text{diag}(\delta_j) \tilde{\lambda}) \\
&= -u_j^T \left( \beta \frac{u_j}{\alpha_j} \right).
\end{aligned} \tag{24}$$

Therefore, we have $\|u_j\|_2 = |\alpha_j|$ and

$$\left\| X^T \tilde{D}_j \tilde{\lambda} + X^T \tilde{S}_j \text{diag}(\delta_j) \tilde{\lambda} \right\|_2 = 1. \tag{25}$$

For the subsampled convex program (13), the KKT conditions are given by: for $i \in \mathcal{I}$, there exists $\zeta^{(i)} \succeq 0$ and $\xi^{(i)}$ such that

$$\begin{aligned}
X^T((T_i)_+ \lambda + T_i \zeta^{(i)} + S_i \xi^{(i)}) + \beta \frac{w_i}{\|w_i\|_2} &= 0, & \text{if } w_i \neq 0, \\
\left\| X^T((T_i)_+ \lambda + T_i \zeta^{(i)} + S_i \xi^{(i)}) \right\|_2 &\leqslant \beta, & w_i = 0, \\
X^T(-(T_i)_+ \lambda + T_i \zeta^{(i)} + S_i \xi^{(i)}) + \beta \frac{w_{i+q}}{\|w_{i+q}\|_2} &= 0, & \text{if } w_{i+q} \neq 0, \\
\left\| X^T(-(T_i)_+ \lambda + T_i \zeta^{(i)} + S_i \xi^{(i)}) \right\|_2 &\leqslant \beta, & \text{if } w_{i+q} = 0.
\end{aligned} \tag{26}$$

Here $S_i$ is a diagonal matrix satisfying that $(S_i)_{jj} = 1$ if $j \in I_0$ and $(S_i)_{jj} = 0$ if $j \in I_+ \cup I_-$, where $\{I_+, I_0, I_-\}$ is the $i$-th trichotomy. The vector $\lambda \in \mathbb{R}^N$ is defined as $\lambda = \nabla \ell \left( \sum_{i \in \mathcal{I}} (T_i)_+ X(w_i - w_{i+q}) \right)$. As $\theta$ is a minimal neural network, there exists a bijective mapping between non-zero neurons $(u_j, \alpha_j)$ and $i \in \mathcal{I}$. For $i \in \mathcal{I}$, suppose that $T_i = \text{diag}(\text{sign}(Xu_j))$. If $\alpha_j > 0$, we let

$$w_i = \alpha_j u_j, w_{i+q} = 0,$$

otherwise, we let

$$w_i = 0, w_{i+q} = -\alpha_j u_j.$$

As the mapping between non-zero neurons $(u_j, \alpha_j)$ and $i \in \mathcal{I}$ is bijective, we note that $\sum_{j=1}^m (Xu_j)_+ \alpha_j = \sum_{i \in \mathcal{I}} X^T \bar{D}_i(w_i - w_{i+q})$. This implies that $\lambda = \tilde{\lambda}$. On the other hand, by taking $\zeta^{(i)} = 0, \xi^{(i)} = \text{diag}(\delta_j) \tilde{\lambda}, \zeta^{(i+q)} = 0, \xi^{(i+q)} = -\text{diag}(\delta_j) \tilde{\lambda}$, as $\tilde{D}_j = (T_i)_+$ and $\tilde{S}_j = S_i$, the KKT conditions (26) are satisfied. Therefore, $W = \{w_i, w_{i+q} | i \in \mathcal{I}\}$ is a global optimum of the subsampled convex program (13).

## B.8   PROOF OF PROPOSITION 5

Let $\tilde{\theta} \in \Theta_m$ be a minimal neural network, and suppose that $W(\tilde{\theta})$ satisfies the KKT conditions of the optimization problem (4). Since the latter is a *convex* optimization problem, it follows that $W(\tilde{\theta})$ is a global minimum. From Proposition 2, we have that $\mathcal{P}^* \leqslant \mathcal{L}_\beta(\theta(W(\tilde{\theta}))) \leqslant \mathcal{L}_\beta^c(W(\tilde{\theta})) = \mathcal{P}_c^* = \mathcal{P}^*$, it follows that $\mathcal{L}_\beta(\theta(W(\tilde{\theta})) = \mathcal{P}^*$ and $\theta(W(\tilde{\theta}))$ is a global minimizer of $\mathcal{L}_\beta$, which yields the claimed result.

### B.9 PROOF OF PROPOSITION 6

Without loss of generality, we can assume that $\theta$ is scaled. Otherwise, we know from the proof of Proposition 3 that $\theta$ can be reduced to a scaled neural network along a continuous path of non-increasing training loss.

We follow the same steps as in the proof of Lemma 1. Denote the neurons of $\theta$ by $(u_1, \alpha_1), \ldots, (u_m, \alpha_m)$. We have that $\widehat{y}(\theta) = \sum_{i=1}^m \widetilde{\lambda}_i z_i$, where $z_i = \text{sign}(\alpha_i) \left( \sum_{j=1}^m \|u_j\| |\alpha_j| \right) \sigma\left( X \frac{u_i}{\|u_i\|} \right)$ and $\widetilde{\lambda}_i = \frac{\|u_i\| |\alpha_i|}{\sum_{j=1}^m \|u_j\| |\alpha_j|}$. Thus, $\widehat{y}(\theta) \in \text{Conv}\{z_1, \ldots, z_m\}$. From Lemma 3, we know that there exist $i_1, \ldots, i_{n+1}$ and $\lambda_1, \ldots, \lambda_{n+1} \geqslant 0$ such that $\sum_{j=1}^{n+1} \lambda_j = 1$ and $\widehat{y}(\theta) = \sum_{j=1}^{n+1} \lambda_j z_{i_j}$. Plugging-in the expressions of the $z_{i_j}$, it follows that

$$
\begin{aligned}
\widehat{y}(\theta) &= \sum_{j=1}^{n+1} \lambda_j \, \text{sign}(\alpha_{i_j}) \left( \sum_{j=1}^m \|u_j\| |\alpha_j| \right) \sigma\left( X \frac{u_{i_j}}{\|u_{i_j}\|} \right) \\
&= \sum_{j=1}^{n+1} \widetilde{\alpha}_{i_j} \sigma(X \widetilde{u}_{i_j}),
\end{aligned}
$$

where

$$
\begin{cases}
\nu_j := \frac{\lambda_j}{\|u_{i_j}\|} \left( \sum_{k=1}^m \|u_k\| |\alpha_k| \right), \\
\widetilde{u}_{i_j} := \sqrt{\frac{\nu_j}{\|u_{i_j}\|}} \, u_{i_j}, \\
\widetilde{\alpha}_{i_j} := \text{sign}(\alpha_{i_j}) \|\widetilde{u}_{i_j}\|.
\end{cases}
$$

Further, we have that

$$
\sum_{j=1}^{n+1} |\widetilde{\alpha}_{i_j}| \|\widetilde{u}_{i_j}\| = \sum_{j=1}^{n+1} \nu_j \|u_{i_j}\| = \sum_{j=1}^{n+1} \lambda_j \left( \sum_{k=1}^m \|u_k\| |\alpha_k| \right) = \sum_{k=1}^m \|u_k\| |\alpha_k|,
$$

where the last equality follows from the fact that $\sum_{j=1}^{n+1} \lambda_j = 1$. Setting $\widetilde{\theta}$ the neural network with neurons $(\widetilde{u}_{i_j}, \widetilde{\alpha}_{i_j})$ for $j = 1, \ldots, n+1$ and $(\widetilde{u}_i, \widetilde{\alpha}_i) = (0, 0)$ for $i \in \{1, \ldots, m\} \setminus \{i_1, \ldots, i_{n+1}\}$, we obtain that $\widetilde{\theta} \in \Theta_m$ and $\mathcal{L}_\beta(\widetilde{\theta}) \leqslant \mathcal{L}_\beta(\theta)$.

Now, we define a continuous path between $\theta$ and $\widetilde{\theta}$, as follows. For $t \in [0, 1]$, $j = 1, \ldots, n+1$ and $i \in \{1, \ldots, m\} \setminus \{i_1, \ldots, i_{n+1}\}$, we set

$$
\begin{aligned}
u_{i_j}(t) &= \frac{(1-t) u_{i_j} |\alpha_{i_j}| + t \, \widetilde{u}_{i_j} |\widetilde{\alpha}_{i_j}|}{\sqrt{\|(1-t) u_{i_j} |\alpha_{i_j}| + t \, \widetilde{u}_{i_j} |\widetilde{\alpha}_{i_j}|\|}} \\
\alpha_{i_j}(t) &= \text{sign}(\alpha_{i_j}) \|u_{i_j}(t)\| \\
u_i(t) &= \sqrt{1-t} \, u_i \\
\alpha_i(t) &= \sqrt{1-t} \, \alpha_i,
\end{aligned}
$$

and $\theta(t)$ the neural network with neurons $\{(u_i(t), \alpha_i(t))\}_{i=1}^m$. Note that $\theta(t)$ is scaled, and

$$
\begin{aligned}
\sum_{i=1}^m \|u_i(t)\| |\alpha_i(t)| &= \sum_{j=1}^{n+1} \|(1-t) u_{i_j} |\alpha_{i_j}| + t \, \widetilde{u}_{i_j} |\widetilde{\alpha}_{i_j}|\| + \sum_{\substack{i=1,\ldots,n+1 \\ i \neq i_1, \ldots, i_{n+1}}} (1-t) \|u_i\| |\alpha_i| \\
&\underset{(i)}{=} (1-t) \sum_{j=1}^{n+1} |\alpha_{i_j}| \|u_{i_j}\| + (1-t) \sum_{\substack{i=1,\ldots,n+1 \\ i \neq i_1, \ldots, i_{n+1}}} \|u_i\| |\alpha_i| + t \sum_{j=1}^{n+1} |\widetilde{\alpha}_{i_j}| \|u_{i_j}\| \\
&\underset{(ii)}{=} (1-t) R(\theta) + t \, R(\widetilde{\theta}) \\
&\underset{(iii)}{=} R(\theta),
\end{aligned}
$$

where equality (i) follows from the triangular inequality and the fact that $\widetilde{u}_{i_j}$ and $u_{i_j}$ are positively colinear; equality (ii) follows from the fact that $\widetilde{\theta}$ and $\theta$ are scaled; equality (iii) holds since $R(\theta) = R(\widetilde{\theta})$. Thus, the function $t \mapsto R(\theta(t))$ is constant over $[0, 1]$.

On the other hand, we have

$$
\widehat{y}(\theta(t)) = \sum_{j=1}^{n+1} \sigma\big(X((1-t)u_{i_j}|\alpha_{i_j}| + t\,\widetilde{u}_{i_j}|\widetilde{\alpha}_{i_j}|)\big)\,\mathrm{sign}(\alpha_{i_j}) + (1-t)\sum_{\substack{i=1,\ldots,n+1\\ i\neq i_1,\ldots,i_{n+1}}} \sigma(Xu_i)\alpha_i
$$

$$
\underset{(i)}{=} (1-t)\sum_{j=1}^{n+1} \sigma(Xu_{i_j})\alpha_{i_j} + t\sum_{j=1}^{n+1} \sigma(X\widetilde{u}_{i_j})\widetilde{\alpha}_{i_j} + (1-t)\sum_{\substack{i=1,\ldots,n+1\\ i\neq i_1,\ldots,i_{n+1}}} \sigma(Xu_i)\alpha_i
$$

$$
= (1-t)\,\widehat{y}(\theta) + t\,\widehat{y}(\widetilde{\theta})
$$

$$
\underset{(ii)}{=} \widehat{y}(\theta)\,,
$$

where equality (i) holds since the $u_{i_j}$ and $\widetilde{u}_{i_j}$ are positively colinear and the $\alpha_{i_j}$ and $\widetilde{\alpha}_{i_j}$ have same signs; equality (ii) holds since $\widehat{y}(\widetilde{\theta}) = \widehat{y}(\theta)$. Consequently, the function $t \mapsto \mathcal{L}_\beta(\theta(t))$ is constant over $[0, 1]$, and this concludes the proof of the fact that $\theta \blacktriangleright \widetilde{\theta}$.

### B.10 Proof of Proposition 7

First, according to Proposition 6, given $\theta \in \Theta_m$ with $m \geqslant n + 1 + m^*$, there exists a neural network $\widetilde{\theta}$ with at most $n + 1$ non-zero neurons such that $\mathcal{L}_\beta(\widetilde{\theta}) \leqslant \mathcal{L}_\beta(\theta)$.

According to Lemma 1, there exists $\theta^* = \{(u_i^*, \alpha_i^*)\}_{i=1}^m$ an optimal neural network with at most $m^*$ non-zero neurons. Up to a permutation of the zero neurons of $\widetilde{\theta}$ and those of $\theta^*$, since $m \geqslant n+1+m^*$, we can assume without loss of generality that $(u_i^*, \alpha_i^*) = (0,0)$ for $i = m^* + 1, \ldots, m$ and $(\widetilde{u}_i, \widetilde{\alpha}_i) = (0,0)$ for $i = 1, \ldots, m^*$.

Now, we define a continuous path between $\widetilde{\theta}$ and $\theta^*$. For $i = 1, \ldots, m^*$ and $j = m^* + 1, \ldots, m$, we set the neural network $\theta(t) \in \Theta_m$ with neurons

$$
\begin{aligned}
u_i(t) &= \sqrt{t}\,u_i^*,\\
\alpha_i(t) &= \sqrt{t}\,\alpha_i^*,\\
u_j(t) &= \sqrt{1-t}\,\widetilde{u}_j,\\
\alpha_j(t) &= \sqrt{1-t}\,\widetilde{\alpha}_j\,.
\end{aligned}
$$

Clearly, we have $\theta(0) = \widetilde{\theta}$ and $\theta(1) = \theta^*$. Further, $\theta(t)$ is scaled and it is easily verified that

$$
\begin{aligned}
R(\theta(t)) &= t\,R(\theta^*) + (1-t)\,R(\widetilde{\theta}),\\
\widehat{y}(\theta(t)) &= t\,\widehat{y}(\theta^*) + (1-t)\,\widehat{y}(\widetilde{\theta})\,.
\end{aligned}
$$

This immediately implies that the function $t \mapsto \mathcal{L}_\beta(\theta(t))$ is convex over $[0, 1]$. Since it achieves a minimum at $t = 1$, it follows that it is non-increasing, and this concludes the proof of Proposition 7.

## C Proofs of intermediate results

### C.1 Proof of Lemma 2

*Proof.* It holds that $\sum_{i\in\mathcal{I}} \sigma(Xu_i)\alpha_i = \sum_{i\in\mathcal{I}} \gamma_i\sigma(Xw_i) = \sum_{i\in\mathcal{I}} D_iXw_i$, whence $\ell(\sum_{i\in\mathcal{I}} \sigma(Xu_i)\alpha_i) = \ell(\sum_{i\in\mathcal{I}} D_iXw_i)$. On the other hand, we have $\|w_{i_j}\|_2 = \|u_j\|_2|\alpha_j|$. Note that $\|u_j\|_2 = |\alpha_j|$ and thus, $\|u_j\|_2|\alpha_j| = \frac{1}{2}(\|u_j\|_2^2 + |\alpha_j|^2)$. Consequently, $\mathcal{L}_\beta(\theta) = \mathcal{L}_\beta^c c(w^*)$. From Pilanci & Ergen (2020), we know that $\mathcal{P}^* = \mathcal{P}_c^*$. Hence, $\mathcal{L}_\beta(\theta) = \mathcal{P}^*$. $\qquad\square$

## C.2 PROOF OF LEMMA 1

We aim to show that $m^* \leqslant n + 1$ and $\mathcal{P}_m^*$ for any $m \geqslant m^*$. We leverage the following result which is known as Caratheodory's theorem.

**Lemma 3.** *Let $z_1, \ldots, z_m \in \mathbb{R}^n$. Suppose that $y \in \mathbf{Conv}\{z_1, \ldots, z_m\}$. Then, there exist indices $i_1, \ldots, i_{n+1} \in \{1, \ldots, m\}$ such that $y \in \mathbf{Conv}\{z_{i_1}, \ldots, z_{i_{n+1}}\}$.*

Suppose that $\theta$ is an optimal neural network with $m \geqslant n + 1$ neurons, and denote its neurons by $(u_1, \alpha_1), \ldots, (u_m, \alpha_m)$. We have that $\widehat{y}(\theta) = \sum_{i=1}^m \widetilde{\lambda}_i z_i$, where $z_i = \text{sign}(\alpha_i) \left( \sum_{j=1}^m \|u_j\| |\alpha_j| \right) \sigma \left( X \frac{u_i}{\|u_i\|} \right)$ and $\widetilde{\lambda}_i = \frac{\|u_i\| |\alpha_i|}{\sum_{j=1}^m \|u_j\| |\alpha_j|}$. Thus, $\widehat{y}(\theta) \in \mathbf{Conv}\{z_1, \ldots, z_m\}$. From Lemma 3, we know that there exist $i_1, \ldots, i_{n+1}$ and $\lambda_1, \ldots, \lambda_{n+1} \geqslant 0$ such that $\sum_{j=1}^{n+1} \lambda_j = 1$ and $\widehat{y}(\theta) = \sum_{j=1}^{n+1} \lambda_j z_{i_j}$. Plugging-in the expressions of the $z_{i_j}$, it follows that

$$\widehat{y}(\theta) = \sum_{j=1}^{n+1} \lambda_j \, \text{sign}(\alpha_{i_j}) (\sum_{j=1}^m \|u_j\| |\alpha_j|) \sigma \left( X \frac{u_{i_j}}{\|u_{i_j}\|} \right)$$

$$= \sum_{j=1}^{n+1} \widetilde{\alpha}_{i_j} \sigma(X \widetilde{u}_{i_j}) \,,$$

where

$$\begin{cases} \nu_j := \frac{\lambda_j}{\|u_{i_j}\|} \left( \sum_{j=1}^m \|u_j\| |\alpha_j| \right) \\ \widetilde{u}_{i_j} := \sqrt{\frac{\nu_j}{\|u_{i_j}\|}} \, u_{i_j} \\ \widetilde{\alpha}_{i_j} := \text{sign}(\alpha_{i_j}) \, \|\widetilde{u}_{i_j}\| \end{cases}$$

Further, we have that

$$\sum_{j=1}^{n+1} |\widetilde{\alpha}_{i_j}| \|\widetilde{u}_{i_j}\| = \sum_{j=1}^{n+1} \nu_j \|u_{i_j}\| = \sum_{j=1}^{n+1} \lambda_j (\sum_{k=1}^m \|u_k\| |\alpha_k|) = \sum_{k=1}^m \|u_k\| |\alpha_k| \,,$$

where the last equality follows from the fact that $\sum_{j=1}^{n+1} \lambda_j$.

We define the neural network $\widetilde{\theta}$ with neurons $(\widetilde{u}_{i_j}, \widetilde{\alpha}_{i_j})$. We have that $\widetilde{\theta} \in \Theta_{n+1}$ and $\mathcal{P}_{n+1}^* \leqslant \mathcal{L}_\beta(\widetilde{\theta}) = \mathcal{L}_\beta(\theta) = \mathcal{P}_m^*$. Since $\mathcal{P}_{n+1}^* \geqslant \mathcal{P}_m^*$ for any $m \geqslant n + 1$, it follows from the previous set of inequalities that $\mathcal{L}_\beta(\widetilde{\theta}) = \mathcal{P}_{n+1}^* = \mathcal{P}_m^*$, and this holds for any $m \geqslant n + 1$. Therefore, $\mathcal{P}_{n+1}^* = \mathcal{P}^*$ and $\mathcal{L}_\beta(\widetilde{\theta}) = \mathcal{P}^*$.

We set $\widetilde{W} = W(\widetilde{\theta})$. We know from Proposition 2 that $W(\widetilde{\theta}) \in \mathcal{W}_{n+1}$ and $\mathcal{L}_\beta^c(\widetilde{W}) \leqslant \mathcal{L}_\beta(\widetilde{\theta})$. Hence, $\mathcal{P}_c^* \leqslant \mathcal{P}^*$. We also know that $\mathcal{L}_\beta(\theta(\widetilde{W})) \leqslant \mathcal{L}_\beta^c(\widetilde{W})$. This implies that $\mathcal{P}_c^* = \mathcal{P}^*$ and $\widetilde{W}$ is an optimal solution to (4). Consequently, $m^* \leqslant n + 1$.

It remains to show that for any $m \geqslant m^*$, we have $\mathcal{P}_m^* = \mathcal{P}^*$. This follows again from Proposition 2. Indeed, let $W^*$ be an optimal solution to (4) such that $W^* \in \mathcal{W}_{m^*}$. Set $\theta^* = \theta(v^*)$. We know that $\theta^* \in \Theta_{m^*}$, and $\mathcal{L}_\beta(\theta^*) \leqslant \mathcal{L}_\beta^c(W^*) = \mathcal{P}_c^* = \mathcal{P}^*$. Hence, $\theta^*$ achieves $\mathcal{P}^*$ and this implies that for any $m \geqslant m^*$, we have $\mathcal{P}_m^* = \mathcal{P}^*$.

## D VERIFICATION OF THE OPTIMAL SET

We review a standard method to determine whether a convex optimization problem has unique solution. Consider a convex optimization problem

$$\min f(x), \text{ s.t. } f_i(x) \leqslant 0, i \in [m], \tag{27}$$

in the variable $x \in \mathbb{R}^d$. Here $f$ and $f_i$ for $i \in [m]$ are convex functions. Suppose that we calculate one optimal solution $x^*$ and the corresponding optimal value $f^*$. We can determine whether $x^*$ is the

unique optimal solution of (27) as follows. For $j \in [d]$, consider the following convex optimization problems

$$p_j^{\mathrm{lb}} = \min x_j, \text{ s.t. } f_i(x) \leqslant 0, i \in [m], f(x) \leqslant f^*, \tag{28}$$

$$p_j^{\mathrm{ub}} = \max x_j, \text{ s.t. } f_i(x) \leqslant 0, i \in [m], f(x) \leqslant f^*. \tag{29}$$

These problems give the upper bound and the lower bound of the value of the $i$-th index in the optimal set of (27). Suppose that $p_j^{\mathrm{ub}} - p_j^{\mathrm{lb}} \leqslant \epsilon$ for certain small $\epsilon > 0$, for instance, $\epsilon = 10^{-8}$. Then, the radius of the optimal set with respect to the $\ell_\infty$ norm is upper-bounded by $\epsilon$. Therefore, we can be confident that $x^*$ is the unique optimal solution up to numerical tolerance.

We have verified numerically that the convex optimization problem in Example 1 in section 3.1 has a unique optimal solution.

