# OpenReview forum: "The Hidden Convex Optimization Landscape of Regularized Two-Layer ReLU Networks: an Exact Characterization of Optimal Solutions"
_ICLR.cc/2022/Conference — ICLR 2022 Oral_

### Official Review · Reviewer_DhNM · 2021-10-31

**Correctness:** 4
**Technical Novelty And Significance:** 4
**Empirical Novelty And Significance:** Not applicable
**Recommendation:** 8
**Confidence:** 3

**Main Review:**

Strengths of the paper:
- The authors provide a complete characterization of the global minima to the nonconvex 2-layer ReLU network training problem, in terms of the solutions of a convex program
- The approach does not rely on duality and/or lifting perspectives of previous convex perspectives on neural network training
- The approach provides an algorithm for testing optimality of a neural network in the studied context
- The work provides significant extensions of the work upon which it is based

Weak points of the paper:
- The method only applies to 2-layer neural networks with ReLU activation and weigh decay.    It is a great contribution that may inspire further developments to weaken these assumptions.

Questions for authors:

One of the technical themes of the paper is the idea of minimal / nearly minimal networks.  Several of the theorems are stated explicitly in terms of nearly minimal networks.  Is it necessary to have minimality in the theorem statements?  That is, could the primary claims be stated without reference to minimality (and where minimality would be a technical detail in the proofs)?


Additional feedback with the aim to improve the paper:

In the model there are no bias terms.  I presume this is because they can be tucked into X with a row of 1's.  Perhaps this is worth a comment in the paper.

Typo: Sec 1.3 "neruons"


**Summary Of The Paper:**

In this paper, the authors study the training of two-layer ReLU networks with weight decay.  A previous paper (Pilanci and Ergen 2020) introduced a convex optimization problem that corresponds to this non-convex case.  In the present paper, the authors prove that all optimal solutions of the nonconvex formulation can be found via optimal solutions of this convex formulation.  (Whereas the Pilanci and Ergen paper only constructed a single solution).  The authors additionally show that a Clarke stationary point of the nonconvex objective corresponds to a global minimum of a subsampled version of the convex problem, and they provide a polynomial time algorithm to test if a neural network is globally optimal.  Finally, the authors prove that the nonconvex loss landscape has no spurious local minima provided the number of neurons is large enough.


**Summary Of The Review:**

Clear accept.  It significantly clarifies the relationship of the solutions of the nonconvex ReLU neural network training problem with a corresponding convex program.  That relationship allows them characterize all global minima and to test whether particular networks are indeed global minima.  While the work only applies in the 2-layer case, it is nonetheless a significant theoretical extension that can inspire extensions to multi-layer cases.

---

> ### Author Response · Authors · 2021-11-17
> **Response to Reviewer DhNM**
>
> For the extension of our results to deep neural networks, please see our separate comment above.
>
> About the question: We believe the subsets of nearly minimal neural network and minimal neural network to be useful in our theorem statements: they provide a simple structure of the set of 2-layer ReLU neural networks and they are directly related to the solutions to the convex program.
>
> Thanks for pointing it out. We add this comment to incorporate the bias term into the model.

---

> > ### Public Comment · ~Claude_Ross1 · 2023-12-04
> > **Response**
> >
> > The international transmission of fiscal policy between trade partners with varied production and indebtedness characteristics https://slopegame.net may be studied in detail using the three-country approach.

---

### Official Review · Reviewer_P5NY · 2021-11-01

**Correctness:** 3
**Technical Novelty And Significance:** 3
**Empirical Novelty And Significance:** 3
**Recommendation:** 8
**Confidence:** 4

**Main Review:**

Strengths:

Interesting observations about and practical ways for constructing the optimal neural network set from the solution of the convex cone program.

The paper is mostly well-written and organized, and technically precise.

The analysis is mostly clean and easy to follow, which may be of interest for addressing related sets of nonconvex problems.

Weaknesses:

The (or similar) regularized convex formulation has been studied in previous works, see e.g., Pilanci & Ergen (2020), Ergen & Pilanci (2021) among several recent others by Pilanci et al. Though a more-in-dept analysis as well as theoretical observations are provided, how technically novel and practical useful of these results shall be further compared and elaborated.

Ergen, Tolga, and Pilanci, Mert. "Convex geometry of two-layer ReLU networks", 2021.

A number of related efforts dealing with learning two-layer ReLU networks by analyzing or modifying the landscape through introducing regularization terms are missing in the discussion; see e.g., Wang et al, 2019. Learning ReLU networks on linearly separable data: Algorithm, optimality, and generalization.
Moreocver, the contributions can be considerably enhanced by providing experimental validations across different tasks such as regression, or classification tasks.

One of the key difficulties in deep learning theory is that learning of deep neural networks is nonconvex due to their compositional structure. How would the results extend to deep neural networks?




**Summary Of The Paper:**

Along the line of Pilanci & Ergen (2020), this submission deals learning two-layer ReLU neural networks through convex optimization. It introduces a number of new notions such as (nearly) minimal neural networks, developing a set of interesting tools, and draws connections between the minimal neural networks and the convex optimization landscape. This paper provides a rich framework along with new analyses and solutions for learning two-layer ReLU networks through convex cone optimization.

**Summary Of The Review:**

The paper contains some typos and grammar errors.

in (1), the second summand should be m terms instead of d terms?

I do not understand why the positive homogeneity of the ReLU activation function directly leads to σ(Xui)αi = σ(Xwi), as \alpha_i's is not necessarily positive in general?

The paragraphs regarding the partions of D_i's are not easy to understand, which shall be improved.


----------------------------------------------------
After rebuttal: I have read the other reviews and authors' replies. My minor issues were addressed. I will keep my score.

---

> ### Author Response · Authors · 2021-11-17
> **Response to Reviewer P5NY**
>
> Thanks for your effort in reviewing our paper and providing many helpful suggestions. We will include the suggestions in the final version, as well as the pointed references. In the remainder, we want to address the main points raised in the reviews.
>
> Compared to [1] and other results related to the convex formulation of regularized training problem, we focus on characterizing the relation between optimal solution to the regularized training problem and the optimal solution to the corresponding convex program. Comparatively, our analysis do not rely on the dual problem, which is the strategy used in [1] and several other works on the convex formulation.
>
> We included the references pointed out about modifying the landscape by introducing the regularization term.
>
> We discussed the extension to deep neural network in another comment above.
>
> For the typos:
>
> 1. Yes, it should be $m$ instead of $d$.
>
> 2. For positive $\alpha$, we have $\sigma(Xu_i\alpha_i)=\sigma(Xw_i)$ and for negative $\alpha$, we have $\sigma(Xu_i\alpha_i)=-\sigma(Xw_i)$. This is the reason why we define the cone $C_{i+p}=C_i$ for $i\in[p]$ and let $D_{i+p}=-D_i$ for $i\in[p]$.
>
> 3. We add more motivations to introduce the partitions of $D_i$ in the revision.
>
> [1] Tolga Ergen, Mert Pilanci, Convex geometry of two-layer ReLU networks.

---

### Official Review · Reviewer_FbD2 · 2021-11-02

**Correctness:** 4
**Technical Novelty And Significance:** 3
**Empirical Novelty And Significance:** 3
**Recommendation:** 8
**Confidence:** 4

**Main Review:**

First of all I have to say this is a very interesting theoretical paper and is the best  among all submissions I have reviewed in ICLR 2022. This paper reveals an interesting hidden convexity structure in  two-layer ReLU neural networks by mapping minimal neural networks to a convex optimization problem and vice versa. The construction of the mapping is clean and elegant. The idea of connecting non-convex loss landscape with convex optimization can potentially be further explored to explain the success of deep learning.  This paper is strong enough and I do not see much weaknesses. But I left to the authors few minor comments which  can be seen as future considerations.

- The first comment that comes to my mind is how the results obtained in the paper can be generalized to neural networks with other activation functions. It seems that the positive homogeneity of ReLU is the key to derive the convex programming problem. Can the authors comment on whether (and how) it is possible to establish similar results for smoother activation functios, e.g. $\text{ReLU}^{k} $ or Sigmoid?

- Can the authors comment on whether and how the results can be extended to deep ReLU neural networks?

Some relevant references are missing.

- Huiyuan Wang, Wei Lin, Harmless Overparametrization in Two-layer Neural Networks, arXiv:2106.04795.

- Quynh Nguyen, On the Proof of Global Convergence of Gradient Descent for Deep ReLU Networks with Linear Widths, arXiv:2101.09612.

- Quynh Nguyen, Pierre Brechet, Marco Mondelli, When Are Solutions Connected in Deep Networks?, arXiv:2102.09671.



**Summary Of The Paper:**

This paper establishes some very interesting connections between the global optimizer of the two-layer ReLU neural networks with the solution of a convex optimization program with cone constraints. More concretely, the authors construct explicit mappings between a smaller class of neural networks called minimal neural networks (that contains all the global optima) and the optimal solutions of convex programming problem. They also show that first-order methods such as SGD can find
networks that can be merged to a minimal representation. Lastly, they provide an explicit path of non-increasing
loss between any point on the loss landscape to the global minimizer. With this, they prove that has no spurious local minima, provided that the number of neurons is sufficiently large.

**Summary Of The Review:**

This paper provides novel tools and insights in revealing the convexity structure of non-convex loss landscape of two-layer neural networks. The ideas developed here can be used to explain the global convergence of gradient descent algorithm in training neural networks. I recommend accepting the paper.

---

> ### Author Response · Authors · 2021-11-17
> **Response to Reviewer FbD2**
>
> Thanks for your effort in reviewing our paper and providing many helpful suggestions. We will include the suggestions in the final version, as well as the pointed references. In the remainder, we want to address the main points raised in the reviews.
>
> 1. Our results heavily depend on the convex optimization formulation of the regularized training problem of two-layer ReLU networks. Going beyond the ReLU activation, there also exist convex optimization formulations for degree two polynomial activations [1,2]. We expect to extend our method to these activation functions in future work. Besides, it is possible to develop convex optimization formulations for the $\mathrm{ReLU}^2$ activation.
>
> 2. Regarding the extension to deep ReLU neural networks, we summarized our response in another comment above.
>
> [1] Burak Bartan and Mert Pilanci, Neural Spectrahedra and Semidefinite Lifts: Global Convex Optimization of Polynomial Activation Neural Networks in Fully Polynomial-Time.
>
> [2] Burak Bartan and Mert Pilanci, Training Quantized Neural Networks to Global Optimality via Semidefinite Programming.

---

### Official Review · Reviewer_VYzk · 2021-11-07

**Correctness:** 3
**Technical Novelty And Significance:** 3
**Empirical Novelty And Significance:** Not applicable
**Recommendation:** 8
**Confidence:** 3

**Main Review:**

Strengths.
I find the results to be novel extensions of of Pilanci and Ergen from 2020 and important theoretically: characterization of the global and local optimum points of the objective function.

Weaknesses.
The paper only considers a single-hidden later. Can this analysis method be extended to multiple layers?


Several typos and corrections I found:
- The upper bound in the rightmost summation in equation (1) should be 'm' not 'd'.
- Page 1, penultimate line,  no definition for $C_i$ for $i\in[p+1, 2p]$ is given.
- Page 3, Section 1.3, first paragraph, third line:
  - Fix the mismatching brace.
  - I believe the $\sigma$ should be dropped.
  - The $1(Xu\geq0)=s$ seems to be a slightly different definition of the cone corresponding to $s$ than the original definition in page 1. Namely, the entries of $Xu$ corresponding to $0$'s in $s$ are required to be negative in this definition and nonpositive in the definition in page 1.
- In equation (5), B(u_j, \alpha_j) has vectors with $d+1$ entries (the last entry comes from $R_{>0}$ or $R_{<0}$ from the definition of  $B_i$) and $C_i$ has vectors with $d$ entries, so the containment under the summation doesn't make much sense here.





**Summary Of The Paper:**

The paper explores the landscape of the objective function in training a single-hidden layer neural network with ReLU activation and L2 regularization. Impressively, the paper has the following contributions:

1. It advances the results of Pilanci and Ergen from 2020 by showing that all *global* optimum points can be found via the convex program introduced by Pilanci and Ergen.
2. It shows that for a large enough width of the hidden layer (at most 2*(n+1), where n is the number of training examples) there are no spurious valleys (i.e . all local minima are global), and GD won't get "stuck".
3. It defines a subclass of single-hidden layer neural networks which it terms "nearly minimal". It characterizes  stationary points of the optimization problem by showing that every such point must be a nearly minimal neural network.
4. It gives a polynomial time algorithm for checking whether a stationary point is a global optimum.


**Summary Of The Review:**

See above.

---

> ### Author Response · Authors · 2021-11-17
> **Response to Reviewer VYzk**
>
> Thanks for your effort in reviewing our paper and providing many helpful suggestions. We will include the suggestions in the final version.
>
> For the extension to deep neural networks, see our official comment.
>
> Thanks for pointing out these typos. We correct them in the revision.
>
> 1. It should be $m$.
>
> 2. The definition shall be $C_{i+p}=C_i$ for $i\in[p]$.
>
> 3. We change our definition of $C_i$ to be the closure of the convex cone of neurons $u \in \mathbb{R}^d$ such that $\mathbf{1}(Xu \geq 0) = s$. This makes the entry of $Xu$ corresponding to $0$s in $s$ to be non-positive.
>
> 4. The cone shall be $B_i$ insead of $C_i$.

---

### Decision · Program_Chairs · 2022-01-20

**Decision:**

Accept (Oral)

**Comment:**

A conceptually and technically highly innovative paper which reinforces an existing powerful connection between the critical set of two-layer ReLU networks and suitable convex programs with cone constraints. The reviewers are in strong consensus that the paper is sound and has merits for publication.